# Physics in 2-Steps: Locking Motion Priors
# Before Visual Refinement Erases Them

Woojung Han [1]   Seil Kang [1]   Youngjun Jun [1]   Min-Hung Chen [2]   Fu-En Yang [2]   Seong Jae Hwang [1]

## Abstract

Image-to-Video diffusion models leverage input images to generate visually stunning content, yet frequently produce motion that violates physical laws. We reveal a surprising finding: a 2-step generation often exhibits *better* physical consistency than a 50-step output from the same model. Through spectral analysis, we trace this to phase erosion during denoising; the phase degrades significantly (dropping by $\approx 18\%$ from step 2 to step 50), whereas the magnitude remains relatively stable. Building on this insight, we propose *PhaseLock*, a training-free framework that preserves the valid motion priors from few-step inference throughout the denoising trajectory. Rather than relying on full-step inference for physical consistency, PhaseLock extracts a motion prior from just 2 steps and enforces it onto high-fidelity generation via *Latent Delta Guidance*. Our approach effectively mitigates phase degradation, improving physical consistency by an average of 6.2 points across diverse models while largely maintaining visual fidelity, with negligible overhead ($1.06\times$ time, $1.02\times$ memory) and reduced reliance on expensive external guidance methods ($\sim 5\times$ time).

## 1. Introduction

Recent advances in video generation have achieved remarkable progress in producing visually coherent and semantically aligned content (Blattmann et al., 2023a; Ho et al., 2022; Brooks et al., 2024). Modern diffusion-based models demonstrate strong capabilities in understanding *what* should appear in a video; objects, scenes, and their visual attributes are rendered with impressive fidelity. Yet, despite this semantic competence, a critical limitation persists: *phys-*

[1]Yonsei University, South Korea [2]NVIDIA, Taiwan. Correspondence to: Seong Jae Hwang <seongjae@yonsei.ac.kr>.

*Proceedings of the 43rd International Conference on Machine Learning*, Seoul, South Korea. PMLR 306, 2026. Copyright 2026 by the author(s).

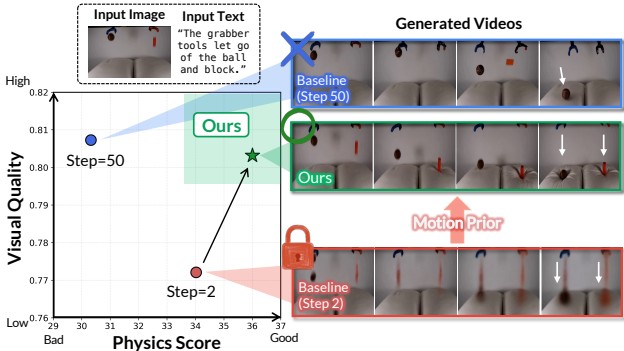

*Figure 1.* **Overview of PhaseLock.** Few-step inference ($T = 2$) captures accurate physical motion (following the white arrow) but lacks textural detail, whereas standard inference ($T = 50$) achieves photorealism but compromises physical integrity with hallucinations. Our method, PhaseLock, extracts the valid motion prior from the few-step inference stage and injects it during the denoising steps, improving physical consistency while largely preserving visual fidelity. Sec. 3.1 details the specific experimental setup.

*ical hallucination*, the generation of motion that violates fundamental physical laws (Kang et al., 2025a; Bansal et al., 2025; Chow et al., 2025). While substantial efforts have been devoted to embedding external physical knowledge through physics engines or external modules (Liu et al., 2024; Yuan et al., 2023; Zhang et al., 2026; Yuan et al., 2026) or scaling model size and data (Yang et al., 2025; Wang et al., 2026), these approaches often demand excessive computational overhead or human annotation. Despite their scale, they continue to produce physically implausible dynamics, impeding the path toward reliable world simulators (Qin et al., 2025). This raises a key question: *does the model lack physical knowledge, or does it forget what it already knows?*

Among video generation settings, Image-to-Video (I2V) provides a controlled lens for studying physical failure: the input image fixes the initial scene, leaving the subsequent motion as the primary degree of freedom for the model. While text prompts can help specify the intended dynamics when the motion is explicit (*e.g.,* "a ball falling"), they often remain ambiguous when they describe only a causal setup (*e.g.,* "letting go of a ball") without specifying the

resulting motion. To address this ambiguity, we turn to the input image, which offers more grounded physical cues than high-level text. Drawing on findings that visual observation supports intuitive physics (Piloto et al., 2022; Gao et al., 2025), prior work suggests that reference images can implicitly encode material properties, structural constraints, and plausible physical states (Liu et al., 2024). Despite the availability of such visual priors, however, we observe that current I2V models still suffer from severe physical hallucinations.

We hypothesize that physical priors encoded in the image fail to propagate due to structural loss during denoising. Motivated by diffusion dynamics studies showing that early denoising steps capture global structure before high-frequency refinement (Song et al., 2020; Qi et al., 2023; Han et al., 2025), we compare few-step and standard inference. We begin with a counter-intuitive observation: a video generated with extremely few denoising steps (*e.g.,* only 2 steps) often exhibits better physical consistency than a standard 50-step generation, as shown in Fig. 1. While the standard 50-step output yields high visual fidelity, it often hallucinates erratic motions or vanishing objects (*e.g.,* failing to capture the correct vertical drop shown in Fig. 1); in contrast, few-step inference better preserves the physical trajectory, as validated by Physics-IQ (Motamed et al., 2026). Since both outputs are generated from the identical model, seed, and conditioning, this divergence reveals a noteworthy trade-off: the model retrieves a valid *"motion prior"* in few-step generation, but can inadvertently *overwrite this physical structure* during visual refinement.

To understand the origin of this degradation, we analyze the generation process in the frequency domain, where magnitude captures appearance-related energy and phase determines the spatial organization of structures across frames. Decomposing the video latent into magnitude and phase components reveals that while the magnitude spectrum remains stable, the phase spectrum degrades significantly (dropping by $\approx 18\%$ from step 2 to step 50), indicating that denoising mainly disrupts structural dynamics rather than appearance energy. We further test this relationship by selectively corrupting either phase or magnitude in GT videos; 50% phase corruption induces $8.5\times$ greater optical flow distortion than equivalent magnitude corruption. Therefore, preserving early phase dynamics enables us to mitigate physics hallucination without training, by effectively leveraging the inherent motion priors.

Based on these findings, we propose **PhaseLock**[1]. Grounded in the observation that physical consistency is established within a few steps, our *Latent Delta Guidance* leverages the few-step latent as a motion prior. Specifically, we employ a straightforward, training-free mechanism that

computes inter-frame deltas from the few-step latent and applies these differences to guide the denoising process. This approach helps preserve phase information during high-fidelity denoising. Our method demonstrates an average improvement of 6.2 points across diverse models. While WMReward (Yuan et al., 2026) shows comparable performance gains, our approach achieves this with much lower computational cost ($1.06\times$ time, $1.02\times$ memory), whereas WMReward requires substantially more resources ($\sim 5\times$ time).

Our contributions are summarized as follows:

- We find that few-step inference yields better physical structure, revealing that the denoising process progressively erodes the phase spectrum (encoding structural dynamics).

- We propose PhaseLock, a model-agnostic, training-free strategy that uses the physical priors captured in few-step inference (NFE = 2) to preserve the structural phase spectrum.

- Our method achieves strong physical consistency (avg. +6.2 points) with marginal overhead ($1.06\times$ time, $1.02\times$ memory), eliminating the need for expensive external guidance ($\sim 5\times$ time).

## 2. Related Works

**Physical Consistency in Video Generation.** Large-scale diffusion models (Ho et al., 2022; Blattmann et al., 2023b; Singer et al., 2023; Bar-Tal et al., 2024; Peebles & Xie, 2023) have achieved remarkable visual quality, yet consistently struggle with physical consistency (Kang et al., 2025a; Bansal et al., 2025; Meng et al., 2025). Benchmarks such as VideoPhy (Bansal et al., 2025) and PhyGenBench (Meng et al., 2025) assess physical plausibility with VLM-based evaluation protocols. Physics-IQ (Motamed et al., 2026) further evaluates physical understanding by comparing whether generated motion, including velocity and trajectory, follows the physically correct evolution from the input image. Several directions have emerged to improve physical consistency (Xue et al., 2025; Zhang et al., 2025). WISA (Wang et al., 2026) curates physics-specific datasets, but dataset-centric approaches can be costly and may still face generalization limits (Kang et al., 2025a). Alternatively, methods like PhysGen (Liu et al., 2024) and VideoREPA (Zhang et al., 2026) incorporate external simulators or foundation-model alignment. More recently, WMReward (Yuan et al., 2026) uses a latent world model as a physics reward for test-time trajectory search and guidance, but incurs substantial overhead.

**Training-Free Inference Strategies.** Training-free methods improve diffusion outputs without retraining through

---

[1] https://dnwjddl.github.io/phaselock/

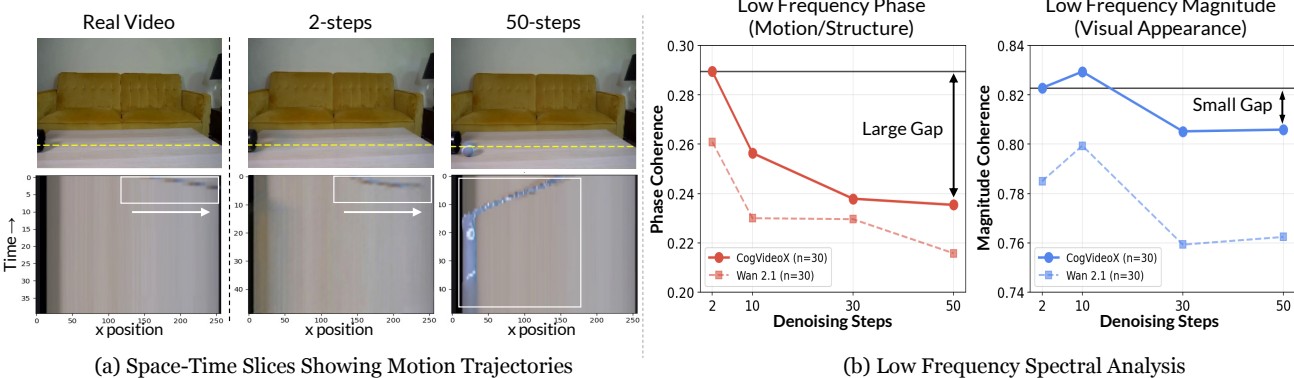

| (a) Space-Time Slices Showing Motion Trajectories | (b) Low Frequency Spectral Analysis |

*Figure 2.* **Analysis of physical degradation across denoising steps.** We compare the baseline models (CogVideoX, Wan 2.1) at few ($T = 2$) and default ($T = 50$) inference steps against the GT video. *(a) Spatio-temporal ($x - t$) slices:* The yellow line indicates the temporal reference axis. As highlighted in the white box, both the GT video and Step 2 accurately follow the physical trajectory. In contrast, Step 50 exhibits severe physical hallucination, moving in the opposite direction. *(b) Low frequency spectral analysis:* We analyze the low-frequency components of Phase and Magnitude across steps $t \in \{2, 10, 30, 50\}$. While magnitude remains consistent between Step 2 and Step 50, phase significantly diverges, suggesting phase corruption as a source of the motion artifacts.

classifier-free guidance (Ho & Salimans, 2022), attention manipulation (Hertz et al., 2022; Hong et al., 2023), and semantic interventions in Diffusion Transformers (Kang et al., 2025b). For video, FreeInit (Wu et al., 2024) refines low-frequency initialization components, while TokenFlow (Geyer et al., 2024) propagates consistent features across frames. However, these methods generally target visual, semantic, or temporal consistency through initialization, attention, or feature propagation, rather than explicitly preserving motion dynamics for physical plausibility.

**Frequency Analysis and Diffusion Dynamics.** Signal processing perspectives have provided key insights into diffusion behavior. Prior studies of diffusion dynamics, including EDM (Karras et al., 2022) and Cold Diffusion (Bansal et al., 2023), characterize generation as a coarse-to-fine process in which global structures form early and high-frequency details emerge later. This perspective has been formalized as spectral autoregression, revealing diffusion models implicitly operate in the frequency domain. Specifically, FreeU (Si et al., 2024) reweights backbone and skip connections by frequency, FreeInit (Wu et al., 2024) refines low-frequency initialization for temporal consistency, and FreqPrior (Yuan et al., 2025) filters noise in the frequency domain. Moving beyond frequency-band analysis, we identify "phase erosion" as a key mechanism associated with physical hallucinations and preserve phase dynamics to improve physical consistency.

## 3. Mechanism Analysis

Here, we systematically analyze the mechanism of physical hallucination through a step-wise examination of video generation. We observe a trade-off between visual refinement and dynamic consistency, and identify phase degradation as

a key mechanism. Further details on the analysis setup are provided in §B.1.

### 3.1. Divergence of Visual and Physical Fidelity

Motivated by the inherent coarse-to-fine generation dynamics of diffusion models (Ho et al., 2022; Choi et al., 2022), where global structure is established before high-frequency details, we hypothesize that physical motion priors may already be captured during few-step denoising. To test this, we compare the generation results at few-step inference ($T = 2$) versus the fully denoised output ($T = 50$). As illustrated in Fig. 1, we observe a noticeable divergence between visual and physical fidelity. Specifically, comparing step 2 and step 50, the visual quality relative to GT videos improves (LPIPS↓ (Zhang et al., 2018): $0.23 \rightarrow 0.19$), yet physical consistency drops (Physics-IQ ↑: $34.02 \rightarrow 30.32$). This quantitative trend aligns with our qualitative observations as shown in Fig. 1. To examine the motion dynamics, we inspect the spatiotemporal ($x$-$t$) slices shown in Fig. 2(a). This visualization reveals that the 2-step result exhibits motion patterns most similar to the GT video. In contrast, the 50-step video suffers from temporal inconsistencies, such as a ball moving backward. Additional visualizations are provided in §B.1.

These results indicate that few-step inference can produce more physically consistent motion than standard denoising, despite lower visual fidelity. Although extending denoising from 50 to 100 steps may seem beneficial, we find that it improves physical consistency by only about 1 point despite the substantial increase in inference time; see §D.2 for details. We therefore ask what information is preserved in the few-step trajectory but weakened during subsequent refinement.

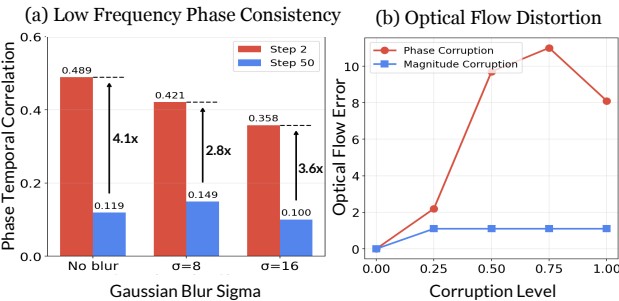

*Figure 3.* **Further analysis on phase properties.** *(a) Blur control:* Even with Gaussian blur applied to match sharpness, Step 2 retains significantly higher phase temporal correlation, indicating that phase loss is structural rather than a frequency artifact. *(b) Phase Sensitivity:* Physical dynamics are highly sensitive to phase corruption, degrading rapidly compared to the stable magnitude.

### 3.2. Spectral Mechanism

To identify the mechanism behind this degradation, we analyze the generation process in the frequency domain. We denote the generated video latent as $z$ and decompose it into magnitude and phase components via the Fourier transform: $\mathcal{F}(z) = A \cdot e^{i\phi}$. Following signal processing theory (Oppenheim, 1999), the phase spectrum $\phi$ encodes spatio-temporal structure (structural layout and motion trajectories), whereas the magnitude spectrum $A$ captures the low-level statistics (texture, contrast).

**Spectral Decomposition.** We analyze the spectral dynamics of CogVideoX (Yang et al., 2025) and Wan 2.1 (Wan et al., 2025), specifically focusing on the low-frequency region (normalized distance $< 0.4$) of the 3D spatio-temporal spectrum. Given that high-frequency details are scarce in the 2-step output, the low-frequency region serves as a reliable indicator of coarse motion dynamics. To quantify spectral fidelity at each denoising step, we employ two metrics: (1) Phase Coherence, defined as the mean cosine similarity between the phase angles of GT and generated video latents, and (2) Magnitude Correlation, measured via the Pearson correlation of their log-magnitudes. As illustrated in Fig. 2(b), while the Magnitude Correlation remains relatively stable with minimal loss (decreasing only by ~2-3%), the Phase Coherence exhibits a sharp degradation, dropping by approximately 18% in both CogVideoX and Wan 2.1 models. These results suggest that the phase degradation observed during refinement is closely tied to the loss of motion dynamics. Spectral analysis results after applying PhaseLock are provided in §D.4.

**Low-Frequency Structural Integrity.** One possible concern is that the high phase consistency of the 2-step output is merely an artifact of its inherent blurriness, which lacks high-frequency disturbances. To rule out this possibility, we applied varying degrees of Gaussian blur ($\sigma \in \{0, 8, 16\}$) to

all outputs (GT, Step 2, and Step 50), progressively filtering out texture, as shown in Fig. 3(a). We then computed the inter-frame phase difference using frame-wise 2D FFT to capture how the temporal phase structure evolves between consecutive frames. The alignment of these dynamics was quantified using the Pearson correlation between the GT and generated phase difference maps. Even under strong blur ($\sigma = 16$), the 2-step output maintained a $3.6\times$ higher phase-difference correlation with the GT video (0.358) than the 50-step output (0.100). This suggests that the phase alignment of the 2-step output reflects genuine structural alignment, rather than merely an artifact of blurriness.

**Phase Sensitivity of Motion Dynamics.** The above analysis shows that phase degradation correlates with physical inconsistency. To further test the role of phase in motion dynamics, we perform controlled corruption experiments on GT videos. Specifically, we decompose video frames via FFT and selectively inject 50% uniform noise into either the phase or magnitude spectrum while keeping the other intact. We then measure motion distortion using optical flow estimated by RAFT (Teed & Deng, 2020), quantified by End-Point Error (EPE), the average Euclidean distance (in pixels) between the original and corrupted flow vectors. As shown in Fig. 3(b), phase corruption induces severe motion distortion (EPE: 9.74, i.e., ~10 pixel average displacement error), whereas equivalent magnitude corruption preserves motion accuracy (EPE: 1.14, ~1 pixel error). This $8.5\times$ disparity provides causal evidence that motion dynamics depend strongly on phase, rather than merely correlating with it. Additional causal studies on other metrics and details of the experimental setup are provided in §B.2. Combined with our observation that denoising erodes phase by 18%, this helps explain why 2-step videos retain better phase information and tend to preserve physical consistency.

**Why Few Steps Preserve Physics.** These three empirical observations, together with the fact that coarse physical motion is largely represented in low-frequency structure (Oppenheim, 1999) and that diffusion models establish such structure early through coarse-to-fine generation, provide a coherent explanation for the low-NFE advantage. The 2-step pass can therefore already encode a reliable estimate of global motion dynamics. Moreover, many video diffusion models are trained with L2-style prediction losses, such as noise, velocity, or flow targets. Under such losses, Parseval's theorem provides a frequency-domain view in which phase errors receive magnitude-weighted gradients, reducing their influence in low-energy regions. This asymmetry is consistent with our observation that phase coherence drops ~18% from step 2 to step 50 while magnitude correlation drops only ~2–3%. Finally, since inter-frame motion is closely reflected in inter-frame phase differences (§C, Eq. 4), this asymmetry can manifest as motion corruption, as supported

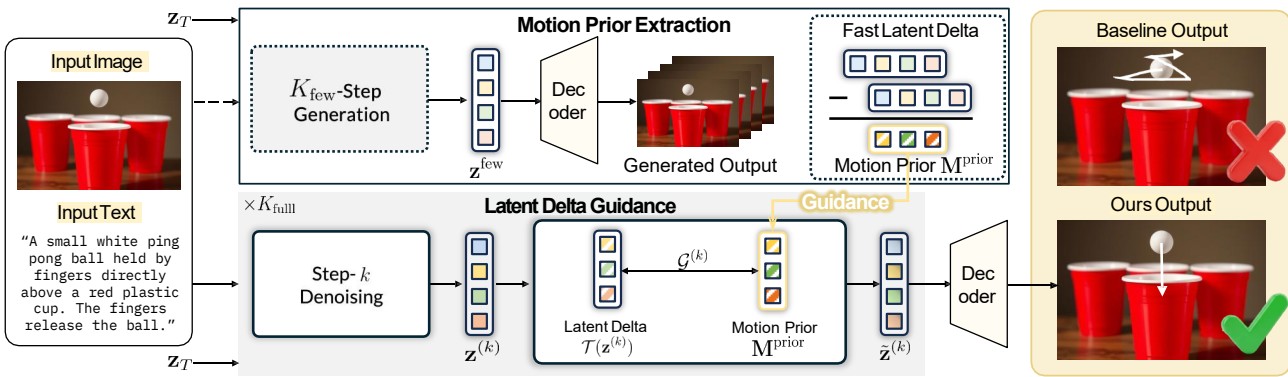

*Figure 4.* **The overall pipeline of PhaseLock.** Our method operates in two distinct stages. *(1) Motion Prior Extraction:* We derive frame-wise motion dynamics from a few-step inference trajectory. *(2) Latent Delta Guidance:* We transfer this motion prior into the standard denoising process. This training-free mechanism effectively enhances physical consistency while preserving high visual fidelity.

by the $8.5\times$ EPE disparity above. Low-NFE inference can therefore preserve physical consistency partly because it limits exposure to the refinement process in which phase is more vulnerable than magnitude. A detailed derivation supporting this explanation is provided in §B.3.

# 4. PhaseLock

Building on our analysis that the few-step prior retains structural phase information, we propose **PhaseLock**, a training-free strategy that guides high-fidelity generation using phase-preserving motion priors. While a direct substitution of the phase spectrum or a selective injection of low-frequency bands might seem intuitive, we avoid such explicit spectral manipulations as they often induce high-frequency artifacts and spatial incoherence (see §D.3). Instead, PhaseLock introduces a spatial-domain proxy for phase preservation by constraining the latent delta, a strategy we theoretically motivate in §C. Our framework operates in two distinct stages: (1) **Motion Prior Extraction**, which obtains a structural guide from few-step inference (Sec. 4.2), and (2) **Latent Delta Guidance**, which aligns high-fidelity generation with the extracted motion priors (Sec. 4.3). A concise summary of the algorithm is available in Alg. 1.

## 4.1. Preliminaries

Let $\mathbf{x} \in \mathbb{R}^{F \times C \times H \times W}$ denote a video sequence with $F$ frames. In Latent Diffusion Models (LDMs), a pre-trained VAE encoder $\mathcal{E}$ maps pixel-space frames to a latent representation $\mathbf{z}_0 = \mathcal{E}(\mathbf{x}) \in \mathbb{R}^{F \times C' \times H' \times W'}$. The diffusion process is defined by a forward chain that progressively adds Gaussian noise, and a reverse chain parameterized by a pre-trained generative model $\boldsymbol{\eta}_\theta(\mathbf{z}_t, t, \mathbf{c})$, where $t$ is the timestep and $\mathbf{c}$ denotes conditioning (*e.g.,* text and image).

**Latent Delta Operator.** We define the Latent Delta Operator $\mathcal{T}$ as the inter-frame difference in the latent space:

$\mathcal{T}(\mathbf{z}) = \mathbf{z}_{2:F} - \mathbf{z}_{1:F-1}$, where $\mathbf{z}_{f:F}$ denotes the temporal slicing of the tensor from frame $f$ to $F$. This operator captures local temporal dynamics while suppressing time-independent features (*e.g.,* static background).

## 4.2. Motion Prior Extraction

Guided by our analysis, we perform a few-step inference process to obtain the motion prior. Given a Gaussian noise initialization $\mathbf{z}_T \sim \mathcal{N}(\mathbf{0}, \mathbf{I})$, we generate a coarse latent sequence $\mathbf{z}^{\text{few}}$ using only $K_{\text{few}}$ denoising steps (*e.g.,* $K_{\text{few}} = 2$): $\mathbf{z}^{\text{few}} = \mathcal{S}(\mathbf{z}_T, \mathbf{c}, K_{\text{few}}; \boldsymbol{\eta}_\theta)$, where $\mathbf{c}$ denotes the conditioning signals (*e.g.,* text and reference image). Here, $\mathcal{S}$ represents the sampling process that maps Gaussian noise to the data manifold using the pre-trained model $\boldsymbol{\eta}_\theta$. Note that we utilize the diffusion backbone $\boldsymbol{\eta}_\theta$ in a frozen state, requiring no additional fine-tuning.

Although latent magnitudes naturally fluctuate during denoising, our empirical analysis in Sec. 3 demonstrates that motion dynamics are relatively robust to magnitude variations but highly sensitive to phase disruptions. This motivates the use of inter-frame latent differences as a phase-sensitive proxy for motion dynamics; we provide theoretical grounding in §C. We extract the motion template $\mathbf{M}^{\text{prior}}$ by applying the latent delta operator to the guide latent:

$$\mathbf{M}^{\text{prior}} = \mathcal{T}(\mathbf{z}^{\text{few}}) = \mathbf{z}^{\text{few}}_{2:F} - \mathbf{z}^{\text{few}}_{1:F-1}. \quad (1)$$

This tensor serves as a structural motion prior that guides the full generation process toward dynamic consistency.

## 4.3. Latent Delta Guidance

In the second stage, we perform the standard high-fidelity generation with $K_{\text{full}}$ steps (*e.g.,* $K_{\text{full}} = 50$). To ensure alignment with the motion prior, we reuse the same initial noise $\mathbf{z}_T$ as in Stage 1. At each denoising step $k$, let $\mathbf{z}^{(k)}$ denote the current intermediate latent. We first compute the

current motion dynamics $\mathbf{M}^{(k)} = \mathcal{T}(\mathbf{z}^{(k)})$. The guidance signal $\mathcal{G}^{(k)}$ is defined as the residual between the target motion prior and the current dynamics, $\mathcal{G}^{(k)} = \mathbf{M}^{\text{prior}} - \mathcal{T}(\mathbf{z}^{(k)})$. This signal captures the deviation of the current trajectory. We inject this guidance specifically into the subsequent frames to align their temporal evolution, while keeping the first frame (anchor) intact:

$$\mathbf{z}_{2:F}^{(k)} \leftarrow \mathbf{z}_{2:F}^{(k)} + \lambda(k) \cdot \mathcal{G}^{(k)}, \qquad (2)$$

where $\lambda(k)$ is a time-dependent scalar controlling the guidance strength. Note that the first frame remains unmodified as it serves as the image condition anchor.

**Adaptive Scheduling.** Since coarse structure is primarily formed in the early phase of diffusion, applying guidance in later steps may interfere with texture refinement. We explicitly decouple motion generation from detail refinement using a linear decay schedule. Let $k \in \{0, 1, \ldots, K_{\text{full}} - 1\}$ denote the elapsed denoising step (*i.e.,* $k = 0$ is the first step from pure noise). The guidance strength is active only during the interval $[k_{\text{start}}, k_{\text{end}})$:

$$\lambda(k) = \begin{cases} \lambda_0 \cdot \left(1 - \frac{k - k_{\text{start}}}{k_{\text{end}} - k_{\text{start}}}\right) & \text{if } k_{\text{start}} \le k < k_{\text{end}} \\ 0 & \text{otherwise.} \end{cases} \qquad (3)$$

This schedule encourages adherence to the motion prior when the global layout is forming and gradually relaxes the constraint to allow for high-fidelity rendering.

### 4.4. Theoretical Justification

Building on our empirical findings, we provide theoretical insight into why constraining latent deltas can help align phase evolution with the motion prior.

**Latent Delta Reflects Phase Differences.** Let $\mathcal{F}$ denote the Fourier transform. We analyze each frequency component separately; for a given frequency, let $A_f e^{j\phi_f}$ denote the corresponding Fourier coefficient of frame $f$, where $A_f$ is the magnitude and $\phi_f$ is the phase. In natural video, consecutive frames typically share similar magnitude spectra, i.e., $A_f \approx A_{f-1} \triangleq A$. Under this assumption, the latent delta $\mathbf{\Delta} = \mathbf{z}_f - \mathbf{z}_{f-1}$ satisfies the following relation, with the full derivation provided in §C:

$$|\mathcal{F}(\mathbf{\Delta})| = 2A \left|\sin\left(\frac{\phi_f - \phi_{f-1}}{2}\right)\right| \approx A \cdot |\phi_f - \phi_{f-1}|, \qquad (4)$$

where the approximation holds for small inter-frame phase shifts ($|\phi_f - \phi_{f-1}| \ll 1$), typical in smooth motion. This reveals that the latent delta magnitude is approximately proportional to the inter-frame phase difference, scaled by the shared magnitude $A$.

*Table 1.* **Comprehensive evaluation on Physics-IQ.** We compare our training-free method against state-of-the-art video generation models. Our approach significantly improves the base models, surpassing much larger proprietary models.

| Type | Model | Params | Score | Gain |
|------|-------|--------|-------|------|
| ***Proprietary Models*** | | | | |
| | Runway Gen-3 Alpha | - | 22.8 | - |
| | VideoPoet | - | 20.3 | - |
| | Lumiere | - | 19.0 | - |
| | Sora | - | 10.0 | - |
| ***Open-Source Models*** | | | | |
| | MAGI-1 | 24B | 30.2 | - |
| | Stable Video Diffusion | 3B | 14.8 | - |
| ***Ours (Plug-and-Play Integration)*** | | | | |
| | CogVideoX (Base) | 5B | 30.8 | - |
| | **+ PhaseLock** | 5B | **36.0** | (+5.2) |
| | LTX-Video (Base) | 2B | 26.4 | - |
| | **+ PhaseLock** | 2B | 32.0 | (+5.6) |
| | Wan 2.1 (Base) | 14B | 20.9 | - |
| | **+ PhaseLock** | 14B | 28.7 | (+7.8) |
| | Wan 2.1 Distilled (LightX2V, 4-step) | 14B | 27.7 | - |
| | **+ PhaseLock** | 14B | 29.4 | (+1.7) |

**Connection to Motion Preservation.** Since the latent delta reflects inter-frame phase differences under the assumptions in Eq. 4, minimizing $\|\mathbf{M}^{\text{prior}} - \mathbf{M}^{(k)}\|$ implicitly constrains phase evolution, which our causal ablation identifies as highly motion-sensitive ($8.5\times$ higher sensitivity than magnitude). Moreover, in normalized latent spaces, magnitude variations are less likely to dominate the loss even when $A_f \approx A_{f-1}$ is imperfect. This spatial-domain approach encourages phase alignment without explicit spectral manipulation that risks artifacts. Further detailed experiments and discussions, including ablation studies using spectral injection, are provided in §D.3.

## 5. Experiments

### 5.1. Setup

**Baselines.** To verify the universality and effectiveness of our proposed method, we conducted experiments on a diverse set of video generation models, ranging from DiT-based architectures to recent open-source large-scale models. Specifically, we utilized CogVideoX-5B (Yang et al., 2025) as our primary baseline. Furthermore, we extended our evaluation to Wan 2.1 (Wan et al., 2025) and LTX-Video (Ha-Cohen et al., 2024) to demonstrate the robustness of our approach across different model architectures.

**Implementation details.** For the inference and evaluation pipeline, we utilized a single NVIDIA H100 (80GB) GPU. We adhered to the default hyperparameters provided by the official repositories of each backbone model unless otherwise specified. Specifically, CogVideoX-5B generates 49 frames at 8 fps with $K_{\text{full}} = 50$ steps; Wan 2.1 produces 81 frames at 16 fps with $K_{\text{full}} = 50$ steps; LTX-Video outputs

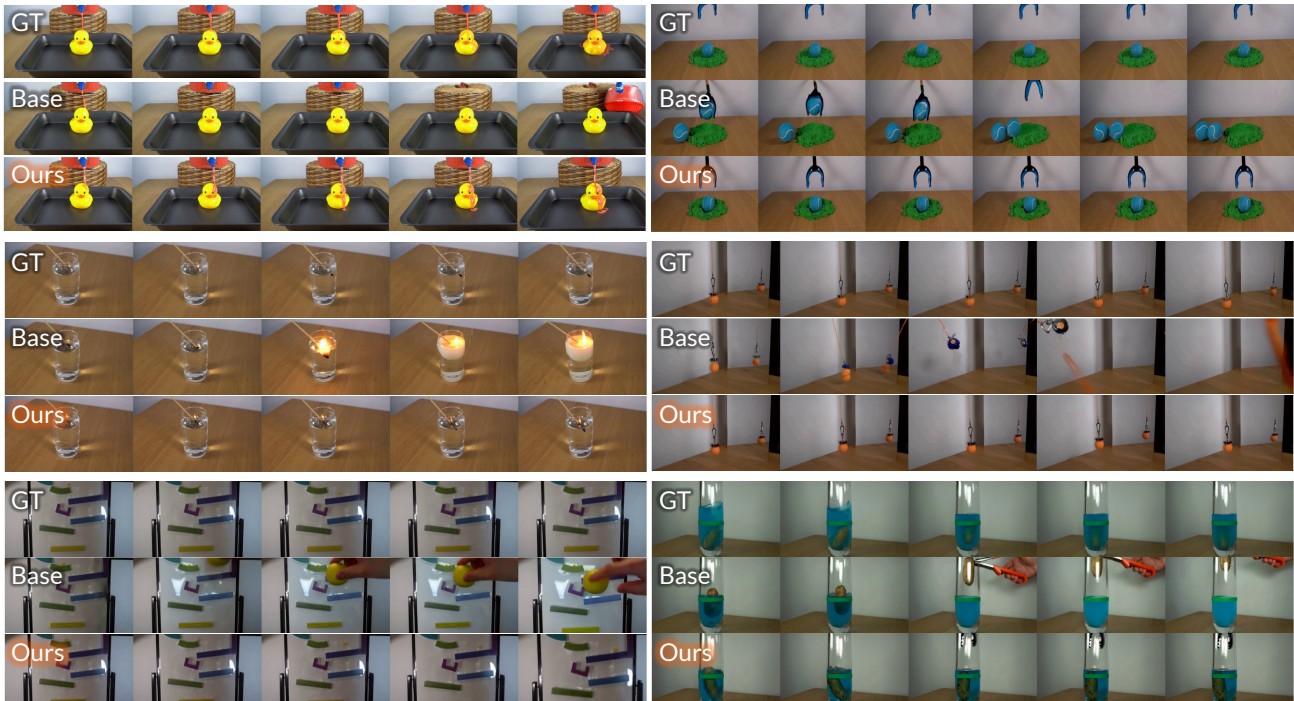

*Figure 5.* **Qualitative results on the Physics-IQ benchmark.** We compare the generated videos from the baseline ('Base') and our method ('Ours'). The results demonstrate that our method exhibits superior adherence to physical laws compared to the baseline, which often fails to maintain physical consistency.

*Table 2.* **Comparison of I2V models across PhyGenBench.** Images are generated using FLUX-schnell, unless marked with [†] (Gemini-2.5 Flash).

| Model | Mech. (↑) | Optics (↑) | Therm. (↑) | Mat. (↑) | Avg. (↑) |
|---|---|---|---|---|---|
| CogVideoX | 0.45 | 0.55 | 0.42 | 0.43 | 0.46 |
| + **PhaseLock** | 0.51 (+13.3%) | 0.78 (+41.8%) | 0.47 (+11.9%) | 0.49 (+14.0%) | 0.57 (+23.9%) |
| CogVideoX[†] | 0.53 | 0.78 | 0.45 | 0.48 | 0.51 |
| + **PhaseLock** | 0.61(+15.1%) | 0.80 (+2.6%) | 0.49 (+8.9%) | 0.52 (+8.3%) | 0.61 (+19.6%) |
| Wan 2.1 | 0.43 | 0.55 | 0.38 | 0.30 | 0.42 |
| + **PhaseLock** | 0.48 (+11.6%) | 0.64 (+16.4%) | 0.49 (+28.9%) | 0.41 (+36.7%) | 0.51 (+21.4%) |
| Wan 2.1[†] | 0.45 | 0.60 | 0.41 | 0.32 | 0.46 |
| + **PhaseLock** | 0.48 (+6.7%) | 0.68 (+13.3%) | 0.50 (+22.0%) | 0.40 (+25.0%) | 0.52 (+13.0%) |

121 frames at 30 fps with $K_{full} = 50$ steps. We applied our guidance mechanism without fine-tuning the pre-trained weights, ensuring a training-free adaptation. In our experiments, we set $\lambda_0 = 0.05$, $k_{start} = 0$, and $k_{end} = K_{full}/2$ (applying a modest initial guidance strength that decays over the first half of denoising).

**Evaluation.** To comprehensively assess the performance of our method, we employed a multi-dimensional evaluation protocol covering both physical plausibility and general video quality. First, we measured physical accuracy using Physics-IQ (Motamed et al., 2026), which objectively calculates the kinematic deviation between generated trajectories and ground-truth videos, and PhyGenBench (Meng et al., 2025), which evaluates perceptual physical plausibility through GPT-4o-based visual reasoning. Addition-

ally, to ensure that the improvement in physical consistency does not compromise the overall visual fidelity, we reported VBench (Huang et al., 2024) scores.

### 5.2. Evaluation Results

**Quantitative Evaluation: Physics-IQ Benchmark.** To rigorously assess physical consistency, we evaluate our method on the Physics-IQ benchmark (Motamed et al., 2026), which offers an objective measure of motion correctness beyond subjective visual metrics. Physics-IQ covers a wide range of physical scenarios, including solid mechanics, fluid dynamics, optics, and magnetism, and measures whether generated motion follows physically correct evolution from the reference image through deviations in object position and velocity. We integrated PhaseLock into diverse video diffusion backbones, including CogVideoX-5B, LTX-Video, and Wan 2.1. As shown in Table 1, Phase-Lock improves standard backbones by an average of 6.2 points, with gains of 5.2, 5.6, and 7.8 points, respectively. PhaseLock also generalizes to step-distilled variants, with a 1.7-point gain on Wan 2.1 (4-step), showing the improvement holds for efficient few-step models and extends across architecturally distinct backbones (§F.2). Since the distilled variant already operates in a few-step regime, its smaller gain is consistent with our explanation that reduced exposure to iterative denoising leaves less phase erosion to correct.

*Table 3.* **Quantitative comparison using VBench.** Using VBench on the Physics-IQ and PhyGenBench benchmarks, we verify that our approach largely preserves visual fidelity while improving physical consistency.

| Model | Subj. Cons. | Back. Cons. | Motion Smooth. | Temp. Flick. | Img. Qual. | Aesth. Qual. |
|---|---|---|---|---|---|---|
| CogVideoX-5B | **0.938** | 0.940 | **0.995** | 0.994 | 0.664 | 0.467 |
| + PhaseLock | 0.935 | **0.955** | **0.995** | **0.995** | **0.680** | **0.489** |
| | (-0.3%) | (+1.5%) | (-0.0%) | (+0.1%) | (+2.3%) | (+4.7%) |
| Wan 2.1 | **0.897** | 0.911 | 0.987 | 0.983 | 0.643 | **0.475** |
| + PhaseLock | 0.881 | **0.921** | **0.995** | **0.993** | **0.655** | 0.451 |
| | (-1.8%) | (+1.0%) | (+0.7%) | (+0.9%) | (+1.8%) | (-5.2%) |

*Table 4.* **Human preference evaluation.** Pairwise comparisons against CogVideoX and Wan 2.1. We report both Win Rate (Win) and Accuracy (Acc) for each model. Ours consistently outperforms the baselines.

| Criteria | Ours vs. CogVideoX | | | | Ours vs. Wan 2.1 | | | |
|---|---|---|---|---|---|---|---|---|
| | Ours | | CogVideoX | | Ours | | Wan 2.1 | |
| | Win | Acc | Win | Acc | Win | Acc | Win | Acc |
| Physics Plausibility | 78.3 | 58.5 | 21.7 | 41.5 | 83.3 | 70.1 | 16.7 | 29.9 |
| Visual Quality | 78.9 | 61.7 | 21.1 | 38.3 | 88.2 | 74.7 | 11.8 | 25.3 |
| Prompt Alignment | 60.4 | 54.0 | 39.6 | 46.0 | 78.5 | 65.8 | 21.5 | 34.2 |

**Quantitative Evaluation: PhyGenBench.** Complementing our Physics-IQ evaluation, we employed PhyGen-Bench (Meng et al., 2025) to assess holistic physical plausibility via Large Vision Language Model (LVLM). To adapt this T2V benchmark for I2V, we utilized input images generated by Gemini-2.5-flash (Comanici et al., 2025) and FLUX-schnell (Esser et al., 2024) (details in §E). Despite the inherent subjectivity of LVLM evaluations, our method demonstrates superior physical robustness, outperforming baselines as shown in Table 2.

**Quantitative Evaluation: VBench.** We further verify that our physics improvements yield VBench scores that remain broadly comparable to the baseline by evaluating generated Physics-IQ and PhyGenBench videos on VBench (Huang et al., 2024) across six visual quality dimensions. As shown in Table 3, PhaseLock keeps most VBench metrics comparable to the baseline while improving several dimensions, such as Background Consistency and Image Quality. While Wan 2.1 shows a modest drop in Aesthetic Quality, the remaining metrics improve or remain comparable, suggesting that the overall visual quality stays at a similar level.

**Quantitative Evaluation: Human Study.** We also supplement our quantitative evaluations with a human study to verify the effectiveness of our approach, following the evaluation protocol in (Yuan et al., 2026). We use all videos from the Physics-IQ benchmark, rather than a curated subset, with a side-by-side comparison interface where 15 annotators view pairs of generated videos. For each video pair,

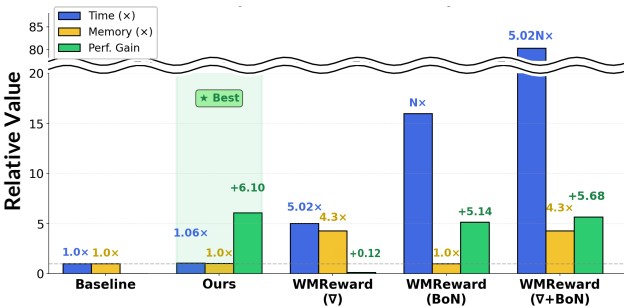

*Figure 6.* **Comparison of Efficiency and Performance.** Note that $N$ denotes the number of generated samples. Our method achieves significant performance gains while maintaining low latency and memory usage comparable to the baseline. In contrast, achieving similar gains with other methods requires substantially higher time and/or memory.

annotators provide judgments across three criteria: Physics Plausibility (whether the motion follows realistic physical dynamics), Visual Quality (overall perceptual quality and clarity), and Prompt Alignment (whether the generated video adheres to the input prompt). Results are aggregated using win rates (excluding neutrals) and accuracy scores to account for ties (wins + 0.5 × neutrals)/total. Table 4 demonstrates that our method delivers significant improvement in all categories. These human judgments complement the LVLM-based PhyGenBench evaluation by directly assessing full videos, and align with our quantitative findings that phase preservation improves physical motion dynamics while maintaining perceptual quality. Further details are provided in §E.

**Qualitative Evaluation.** Our method demonstrates the capability to synthesize dynamic elements as shown in Fig. 5. Compared to the baseline, our results are significantly more stable, effectively avoiding artifacts such as motion jitter and the random appearance of new objects. Specific examples highlight this physical consistency. In the bottom-right of Fig. 5, the liquid level naturally rises upon object entry, consistent with Archimedes' principle. In the bottom-left example, while the baseline fails to preserve small objects, causing the ball to vanish, our model maintains a smooth, continuous trajectory. See §F.4 for more results.

### 5.3. Efficiency and Test-Time Cost

We analyze the test-time cost of PhaseLock on CogVideoX-5B and compare it with WMReward (Yuan et al., 2026), a state-of-the-art training-free physics-alignment method on Physics-IQ. As shown in Fig. 6, PhaseLock remains competitive with WMReward (36.0 vs. 36.3 on Physics-IQ) while incurring only a $1.06\times$ inference-time overhead, primarily from the additional few-step pass used to extract the motion prior. In contrast, WMReward relies on VJEPA-2 (Assran et al., 2025) as an external world-model reward for Best-

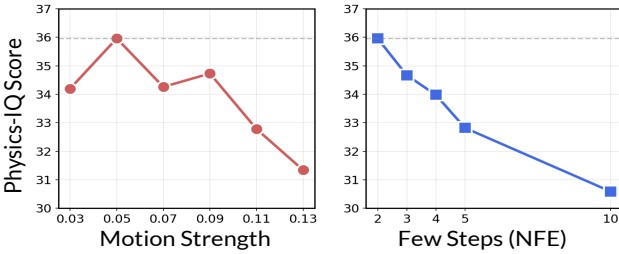

*Figure 7.* **Ablation studies on hyperparameters.** Impact of motion strength and the number of few-step inference steps ($K_{\text{fast}}$, NFE) on Physics-IQ scores. Performance peaks at strength $0.05$ and at NFE $= 2$.

of-N search and gradient-based guidance. The subsequent Latent Delta Guidance consists of lightweight latent tensor operations, adding negligible cost relative to the diffusion backbone. Thus, PhaseLock approaches the performance of reward-based test-time scaling while avoiding external reward models, gradient backpropagation, and expensive Best-of-N search. For a broader comparison with other physics-guided video generation methods, Table 18 summarizes the trade-offs in performance, cost, and applicability.

### 5.4. Ablation Studies

To assess the sensitivity of our model to key hyperparameters, we conducted ablation studies on motion strength $\lambda_0$ and the number of few steps (NFE). First, regarding the motion strength, we varied the value from 0.03 to 0.13. As shown in Fig. 7, a strength of 0.05 yields the highest Physics-IQ score, whereas higher values lead to a decline in performance. Second, for the few-step inference, we evaluated NFE values ranging from 2 to 10. We observed that performance is maximized at NFE = 2, with a consistent decrease in scores as the number of steps increases. More ablation studies on scheduling, guidance formulation, and hyperparameters are provided in §F.1.

### 6. Discussion

PhaseLock is most beneficial for scenarios where preserving coherent inter-frame motion is critical. Physics-IQ covers diverse physical phenomena, including liquids, thermodynamics, magnetism, deformable objects, and rigid-body motion; per-scenario score gains are analyzed in §F.3. Because Latent Delta Guidance constrains the inter-frame velocity field in the spatial domain, sharp-boundary and high-frequency events can also benefit. The smaller gain on the step-distilled Wan 2.1 variant is also expected, as it is already optimized for few-step generation and leaves less denoising-induced phase erosion for PhaseLock to correct.

**Limitations.** Because PhaseLock transfers whatever motion is captured by the 2-step pass rather than injecting

external physical knowledge, a physically implausible few-step prior propagates to the final output, and this cannot reliably be corrected by tuning $\lambda_0$. We do not assume that the few-step prior is universally correct; instead, our per-scenario analysis characterizes unreliable priors through the cases where guidance degrades performance. The failure is scenario-specific rather than global: PhaseLock improves 74% of Physics-IQ scenarios on Wan 2.1 and 67% on CogVideoX-5b, and the largest per-scenario degradation is smaller than half of the largest per-scenario gain. The method also assumes an iterative denoising loop and is therefore not directly applicable to autoregressive video generators. A detailed per-scenario breakdown and failure categorization are provided in §F.3 and §G, respectively.

**Future Directions.** Our analysis suggests that phase erosion may arise from the magnitude-weighted behavior of common MSE-style training objectives, motivating phase-aware training losses as a complementary training-time remedy. PhaseLock could also be combined with training-based physics methods to compound their respective gains. In parallel, adaptive guidance mechanisms could adjust the strength of PhaseLock based on the reliability of the few-step prior. More broadly, the observation that coarse generations can preserve dynamics lost during refinement may extend to autoregressive video generators.

### 7. Conclusion

We reveal that video diffusion models establish physically consistent motion within just 2 steps, yet progressively overwrite this knowledge during visual refinement. PhaseLock locks these motion priors before they are lost, improving physical consistency by 6.2 points on average at negligible cost ($1.06\times$ time) without external guidance or additional training. Our findings suggest that physics-aware generation may require not more computation, but smarter preservation, opening directions toward phase, preserving samplers, physics-aware training objectives, and extensions to audio and 3D domains.

### Impact Statement

Our research investigates the internal mechanisms of diffusion models to mitigate physical hallucinations in video generation. By ensuring generated content adheres to physical laws, this work contributes to applications requiring high reliability, such as robotics simulation and scientific visualization. We acknowledge that advancing generative capabilities implies potential societal risks, including the creation of misleading media. However, we believe that understanding the limitations and mechanisms of these models is essential for building more transparent, controllable, and physically grounded AI systems.

## Acknowledgements

This work was supported in part by the IITP RS-2024-00457882 (AI Research Hub Project), IITP 2020-II201361, NRF RS-2024-00345806, NRF RS-2023-002620, NRF-2024S1A5C3A03046579, and RQT-25-120390. Affiliations: Department of Artificial Intelligence (Y.J, S.J.H), Department of Computer Science (W.H, S.K).

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

# Technical Appendices

## A. Additional Material: Media Gallery

As displaying video content frame by frame within the paper offers only limited insight into temporal coherence and visual quality, we provide a Media Gallery Page featuring the full video outputs from both the main and additional experiments. This page allows for a more faithful assessment of motion consistency and prompt alignment. The results can be viewed at: https://phaselock-physical-video.github.io/

## B. Analysis Details

### B.1. Setup Details

To conduct the analysis for our main paper, we utilized the Physics-IQ benchmark (Motamed et al., 2026), which provides Ground Truth (GT) videos. We employed Diffusion Transformer models, specifically CogVideoX (Yang et al., 2025) and Wan 2.1 (Wan et al., 2025). The mechanistic analyses in Sec. 3 were performed using 30 to 50 randomly selected paired samples, while the main Physics-IQ benchmark results in Sec. 5 use all 396 videos. Our analysis and method rely on the denoising process; therefore, autoregressive Transformer-based models such as VideoPoet (Kondratyuk et al., 2024) and MAGI-1 (Teng et al., 2025) are not applicable to this study. All experiments were conducted on NVIDIA H100 (80GB) GPUs. Unless explicitly stated otherwise, we adopted the default hyperparameters provided by the official repositories. We applied the benchmark settings, using identical input modalities, including text prompts and reference images.

**Divergence of Visual and Physical Fidelity.** In Fig 1 at main paper, we present a quantitative comparison between the baseline (at denoising steps $t = \{2, 50\}$) and our proposed method. We utilized the comprehensive Physics-IQ benchmark to objectively measure physical consistency, specifically quantifying frame-by-frame motion dynamics (Physics Score). For visual fidelity, we employed the LPIPS metric (Zhang et al., 2018). To facilitate an intuitive visualization where higher values indicate better performance for both axes, we report visual quality as '$1 - \text{LPIPS}$'. Complementing the figure, Table 5 provides the detailed numerical results, including additional evaluations for the baseline at intermediate steps ($t = 10, 30$).

*Table 5.* **Quantitative comparison on the Physics-IQ benchmark.** We evaluate the physical consistency and visual quality across different denoising steps of the baseline and our method PHASELOCK. Note that Visual Quality is reported as ($1 - \text{LPIPS}$) so that higher values indicate better quality for both metrics.

| Method | Physics-IQ ($\uparrow$) | Visual Quality ($1 - \text{LPIPS}, \uparrow$) |
|---|---|---|
| CogVideoX (Step 2) | 34.02 | 0.7721 |
| CogVideoX (Step 10) | 32.84 | 0.8183 |
| CogVideoX (Step 30) | 31.76 | 0.8022 |
| CogVideoX (Step 50) | 30.82 | 0.8073 |
| **Ours** | **36.00** | 0.8020 |

**Spatio-Temporal Slicing.** To explicitly analyze the motion trajectories, we employed spatio-temporal slicing (often referred to as $x$-$t$ slices). Specifically, we defined a cross-section indicated by a yellow reference line on the video frames and extracted the temporal evolution of pixels along this axis. We visualized the trajectories at Step 2 and Step 50 using samples from the Physics-IQ benchmark in Fig. 16, Fig. 17, and Fig. 18. This visualization allows us to intuitively assess the continuity and physical plausibility of the generated motion across different inference steps.

### B.2. Causal Studies

To test the hypothesis that phase information carries structural cues important for physical dynamics, we conducted a controlled causal study utilizing CogVideoX as a representative model. We systematically injected synthetic noise into the phase and magnitude spectra, varying the corruption level from $\alpha = 0$ (original) to $\alpha = 1$ (fully corrupted) for each component separately and quantified the resulting physical degradation across three hierarchical levels of motion derivatives: (1) object trajectory (position), (2) velocity profile (speed), and (3) motion smoothness (jerk). All quantitative evaluations

were performed by comparing the generated outputs against the GT videos.

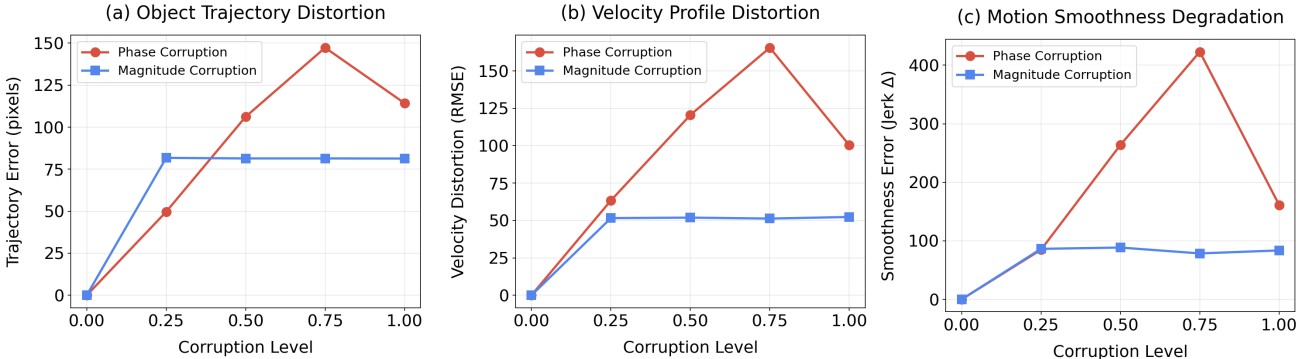

*Figure 8.* **Causal analysis of phase and magnitude corruption on physical dynamics.** We compare the impact of spectral corruption ($\alpha$) across three hierarchical metrics. *(a) Trajectory Error (Position):* Phase corruption causes continuous drift in object location, while magnitude error saturates at $\alpha = 0.25$. *(b) Velocity Distortion (Speed):* Phase corruption severely impacts velocity consistency compared to the invariant response of magnitude. *(c) Motion Smoothness (Jerk):* Phase noise induces physically impossible, jerky motion, showing explosive error growth in higher-order derivatives.

**(1) Object Trajectory Error.** This metric measures the positional deviation of the object, answering "Is the object where it is supposed to be?" We extract the object centroid $\mathbf{c} = (c_x, c_y)$ using image moments $M_{pq}$ derived from Canny edge maps:

$$c_x = \frac{M_{10}}{M_{00}}, \quad c_y = \frac{M_{01}}{M_{00}} \tag{5}$$

The trajectory error is defined as the Euclidean distance between the original and corrupted centroids over time $T$:

$$E_{\text{traj}} = \frac{1}{T} \sum_{t=1}^{T} \|\mathbf{p}_t^{\text{orig}} - \mathbf{p}_t^{\text{corr}}\|_2 \tag{6}$$

As shown in Fig. 8(a), magnitude corruption leads to an immediate saturation in error ($\approx 81$px at $\alpha = 0.25$) but does not degrade further, as the structural edges remain intact despite visual artifacts (*e.g.,* blurring or contrast shifts). In contrast, phase corruption causes a progressive increase in error, eventually surpassing magnitude error at $\alpha \geq 0.5$. This supports the view that phase contributes strongly to the structural location of the object.

**(2) Velocity Profile Distortion.** This metric evaluates whether the object follows physical laws of motion, such as the linear velocity increase in free-fall ($v \propto t$). We compute the velocity sequence $v_t = \|\mathbf{p}_{t+1} - \mathbf{p}_t\|_2$ and measure the Root Mean Square Error (RMSE):

$$E_{\text{vel}} = \sqrt{\frac{1}{T-1} \sum_{t=1}^{T-1} (v_t^{\text{orig}} - v_t^{\text{corr}})^2} \tag{7}$$

As shown in Fig. 8 (b), magnitude corruption shows negligible impact on velocity profiles, with error remaining constant regardless of noise intensity. However, phase corruption induces severe distortion, resulting in $2.32\times$ higher error than magnitude at $\alpha = 0.5$. Since velocity is the first derivative of position ($v = d\mathbf{p}/dt$), small positional jitters caused by phase noise are amplified, disrupting the linear consistency required for physical realism (*e.g.,* constant acceleration).

**(3) Motion Smoothness Error.** To assess the naturalness of the motion, we analyze the Jerk (the rate of change of acceleration). In natural physics (*e.g.,* gravity), acceleration is constant ($\mathbf{a} \approx g$), implying that jerk should be minimal ($\mathbf{j} \approx 0$). We define the jerk vector $\mathbf{j}_t$ and the smoothness error $E_{\text{smooth}}$ as:

$$\mathbf{j}_t = \mathbf{p}_{t+3} - 3\mathbf{p}_{t+2} + 3\mathbf{p}_{t+1} - \mathbf{p}_t, \quad E_{\text{smooth}} = |\bar{J}^{\text{corr}} - \bar{J}^{\text{orig}}| \tag{8}$$

The impact of phase corruption becomes most pronounced in high-order derivatives. While magnitude corruption has almost no effect on smoothness (ratio $\approx 1.0$), phase corruption causes an explosive increase in jerk error, reaching $5.39\times$ the error of magnitude at $\alpha = 0.75$ as shown in Fig. 8(c). This demonstrates that phase noise manifests as physically impossible, jerky motion.

## B.3. Why Low-NFE Exhibits Better Physics

In the main text (Sec. 3) we summarize the mechanism behind the low-NFE advantage. Here we provide the full derivation in three steps.

**Step 1: Physical Motion Is Captured at Low NFE.**   Diffusion models generate content in a coarse-to-fine manner: global low-frequency structure is resolved in early denoising steps, while fine-grained high-frequency detail emerges later. This perspective has been formalized as spectral autoregression in the reverse process. Since physical motion (object trajectories, velocity profiles, momentum transfer) is predominantly a low-frequency phenomenon in the spatial spectrum, the 2-step output already encodes the model's best estimate of global motion dynamics. The physical motion prior is thus not missing; it is the dominant low-frequency structure established in the earliest steps.

**Step 2: Phase Is More Vulnerable Than Magnitude Under L2 Training.**   Many video diffusion formulations use L2-style prediction objectives, such as noise, velocity, or flow targets. Under such objectives, Parseval's theorem gives a frequency-domain view in which the pixel/latent-domain loss decomposes into per-frequency contributions. For a single frequency with ground-truth coefficient $F = |F|e^{j\phi}$ and prediction $\hat{F} = |\hat{F}|e^{j\hat{\phi}}$, the per-frequency squared error $|\hat{F} - F|^2$ yields gradients:

$$\frac{\partial \mathcal{L}}{\partial |\hat{F}|} \propto \left( |\hat{F}| - |F|\cos(\hat{\phi} - \phi) \right), \tag{9}$$

$$\frac{\partial \mathcal{L}}{\partial \hat{\phi}} \propto |\hat{F}| \cdot |F| \cdot \sin(\hat{\phi} - \phi). \tag{10}$$

The phase gradient is modulated by $|\hat{F}| \cdot |F|$, and therefore vanishes wherever magnitude is small, precisely where subtle motion lives in the high-frequency bands. The network is consequently trained with structurally weaker phase-correction than magnitude-correction capabilities, and the asymmetry grows with frequency. While this analysis describes the loss landscape rather than inference dynamics directly (the network outputs pixel/latent values, not $|\hat{F}|$ and $\hat{\phi}$ independently), the asymmetry shapes what the model learns to preserve. Empirical measurements are consistent with this prediction: phase coherence drops by approximately $18\%$ from step 2 to step 50 (Sec. 3, Fig. 2b), while magnitude correlation drops by only $\sim 2$–$3\%$.

**Step 3: Phase Degradation Directly Manifests as Motion Corruption.**   The analysis in §C (Eq. 4) shows that the magnitude of the latent delta is proportional to the inter-frame phase difference, i.e., inter-frame motion is encoded as inter-frame phase difference in the Fourier domain. The phase vulnerability established in Step 2 can therefore propagate into corrupted motion dynamics. Our causal ablation in Sec. 3 supports this link quantitatively: injecting $50\%$ uniform noise into the phase spectrum induces an $8.5\times$ larger optical flow End-Point Error than equivalent noise injected into the magnitude spectrum (EPE: 9.74 vs 1.14). **Conclusion.** Taken together, Steps 1–3 explain the low-NFE advantage as a

structural rather than accidental property: low-NFE inference preserves physics *not despite* fewer steps, *but because* it limits exposure to the denoising process in which phase information relevant to motion is more vulnerable than magnitude. This is also why explicit frequency-domain manipulation alone does not match our spatial-domain proxy (see §D.3): the spatial L2 constraint on the latent delta aggregates phase alignment across all frequencies simultaneously via Parseval's theorem, rather than re-introducing the very asymmetry we are trying to avoid.

# C. Proof of Latent Delta Encoding Phase Differences

We provide the full mathematical derivation establishing that the latent delta encodes inter-frame phase differences. This proof underlies the theoretical claim referenced in the main text (Sec. 3).

We emphasize that the equal-magnitude assumption invoked below ($A_f \approx A_{f-1}$) is a *sufficient* condition for the closed-form in Eq. 4, not a necessary condition for PhaseLock. The general-case derivation in Sec. C.4 shows that the inter-frame phase difference remains an important factor determining $|\mathcal{F}(\boldsymbol{\Delta})|$ even when magnitudes differ. Moreover, our spatial-domain L2 constraint aggregates phase alignment across all frequencies simultaneously via Parseval's theorem, so no single frequency needs to satisfy the equal-magnitude condition exactly.

## C.1. Setup and Notation

Let $\mathcal{F}$ denote the Fourier transform. For a given frequency component, let the Fourier coefficients of consecutive frames $f-1$ and $f$ be:

$$\mathcal{F}(z_{f-1}) = A_{f-1} \cdot e^{j\phi_{f-1}}, \tag{11}$$

$$\mathcal{F}(z_f) = A_f \cdot e^{j\phi_f}, \tag{12}$$

where $A_f, A_{f-1} \in \mathbb{R}^+$ are magnitudes and $\phi_f, \phi_{f-1} \in [-\pi, \pi)$ are phases.

## C.2. Derivation

*Proof.* The latent delta is defined as $\boldsymbol{\Delta} = z_f - z_{f-1}$. By linearity of the Fourier transform:

$$\mathcal{F}(\boldsymbol{\Delta}) = \mathcal{F}(z_f) - \mathcal{F}(z_{f-1}) = A_f e^{j\phi_f} - A_{f-1} e^{j\phi_{f-1}}. \tag{13}$$

**Step 1: Magnitude Stability Assumption.** In natural video, consecutive frames exhibit similar magnitude spectra due to temporal continuity:

$$A_f \approx A_{f-1} \triangleq A. \tag{14}$$

Under this assumption:

$$\mathcal{F}(\boldsymbol{\Delta}) = A \left( e^{j\phi_f} - e^{j\phi_{f-1}} \right). \tag{15}$$

**Step 2: Complex Exponential Difference.** We simplify $e^{j\phi_f} - e^{j\phi_{f-1}}$ using the identity for difference of complex exponentials. Define:

$$\bar{\phi} = \frac{\phi_f + \phi_{f-1}}{2}, \quad \Delta\phi = \phi_f - \phi_{f-1}. \tag{16}$$

Then $\phi_f = \bar{\phi} + \frac{\Delta\phi}{2}$ and $\phi_{f-1} = \bar{\phi} - \frac{\Delta\phi}{2}$. Substituting:

$$e^{j\phi_f} - e^{j\phi_{f-1}} = e^{j(\bar{\phi}+\Delta\phi/2)} - e^{j(\bar{\phi}-\Delta\phi/2)} \tag{17}$$

$$= e^{j\bar{\phi}} \left( e^{j\Delta\phi/2} - e^{-j\Delta\phi/2} \right). \tag{18}$$

**Step 3: Euler's Formula.** Using Euler's formula, $e^{jx} - e^{-jx} = 2j\sin(x)$:

$$e^{j\phi_f} - e^{j\phi_{f-1}} = e^{j\bar{\phi}} \cdot 2j \sin\left( \frac{\Delta\phi}{2} \right). \tag{19}$$

**Step 4: Taking the Magnitude.** Since $|e^{j\bar{\phi}}| = 1$ and $|j| = 1$:

$$\left| e^{j\phi_f} - e^{j\phi_{f-1}} \right| = 2 \left| \sin\left( \frac{\Delta\phi}{2} \right) \right| = 2 \left| \sin\left( \frac{\phi_f - \phi_{f-1}}{2} \right) \right|. \tag{20}$$

Therefore:

$$\boxed{|\mathcal{F}(\boldsymbol{\Delta})| = 2A \left| \sin\left( \frac{\phi_f - \phi_{f-1}}{2} \right) \right|.} \tag{21}$$

**Step 5: Small Angle Approximation.** For smooth motion, inter-frame phase shifts are small: $|\phi_f - \phi_{f-1}| \ll 1$. Using Taylor expansion $\sin(x) \approx x$ for $|x| \ll 1$:

$$|\mathcal{F}(\boldsymbol{\Delta})| \approx 2A \cdot \frac{|\phi_f - \phi_{f-1}|}{2} = A \cdot |\phi_f - \phi_{f-1}|. \tag{22}$$

$\square$

### C.3. Interpretation

This result establishes that:

$$|\mathcal{F}(\boldsymbol{\Delta})| \propto |\phi_f - \phi_{f-1}|, \tag{23}$$

i.e., the latent delta magnitude is directly proportional to the inter-frame phase difference. Consequently, minimizing the latent delta error $\|\mathbf{M}^{\text{prior}} - \mathbf{M}^{(k)}\|$ in the spatial domain effectively constrains the phase evolution in the frequency domain, aligning it to the motion prior.

### C.4. General Case: Unequal Magnitudes

When the magnitude stability assumption does not hold exactly ($A_f \neq A_{f-1}$), we derive the general expression.

*Proof.* Starting from:

$$\mathcal{F}(\boldsymbol{\Delta}) = A_f e^{j\phi_f} - A_{f-1} e^{j\phi_{f-1}}. \tag{24}$$

The squared magnitude is:

$$|\mathcal{F}(\Delta)|^2 = \mathcal{F}(\Delta) \cdot \mathcal{F}(\Delta)^* \tag{25}$$

$$= \left(A_f e^{j\phi_f} - A_{f-1} e^{j\phi_{f-1}}\right)\left(A_f e^{-j\phi_f} - A_{f-1} e^{-j\phi_{f-1}}\right) \tag{26}$$

$$= A_f^2 + A_{f-1}^2 - A_f A_{f-1} e^{j(\phi_f - \phi_{f-1})} - A_f A_{f-1} e^{-j(\phi_f - \phi_{f-1})} \tag{27}$$

$$= A_f^2 + A_{f-1}^2 - 2A_f A_{f-1} \cos(\phi_f - \phi_{f-1}). \tag{28}$$

Therefore:

$$\boxed{|\mathcal{F}(\boldsymbol{\Delta})|^2 = A_f^2 + A_{f-1}^2 - 2A_f A_{f-1} \cos(\phi_f - \phi_{f-1}).} \tag{29}$$

$\square$

This is the law of cosines for complex vectors. The phase difference $(\phi_f - \phi_{f-1})$ controls the cosine term, which provides an important modulation. Even with magnitude variations, the phase difference remains a key factor determining $|\mathcal{F}(\boldsymbol{\Delta})|$.

When $A_f = A_{f-1} = A$:

$$|\mathcal{F}(\boldsymbol{\Delta})|^2 = 2A^2(1 - \cos(\phi_f - \phi_{f-1})) = 4A^2 \sin^2\left(\frac{\phi_f - \phi_{f-1}}{2}\right), \tag{30}$$

which recovers our earlier result.

## D. Methods Details

### D.1. Algorithm of PhaseLock

We present the detailed inference pseudocode for PhaseLock in Algorithm 1. The process consists of two stages: (1) extracting a coarse motion prior ($\mathbf{M}^{\text{prior}}$) using a few-step sampling strategy, and (2) guiding the full generation process to align with this prior via a linearly decaying guidance strength $\lambda$.

### D.2. Impact of Inference Steps on Physical Consistency

We conducted a step-wise analysis using the CogVideoX-5b (I2V) model to address concerns that our results might be attributed to additional processing steps. Although higher NFEs (Number of Function Evaluations) are often associated with better quality, our experiments show that this does not guarantee improved physical accuracy and comes with high computational overhead. Table 6 illustrates that even when the number of timesteps is doubled, the model shows only a marginal gain of about 1 point, remaining significantly lower than our method's performance of 36.0.

---

**Algorithm 1** PHASELOCK

---

**Require:** Reference Image $\mathbf{x}_{\text{ref}}$, Text Prompt $p$, Pre-trained Diffusion Model $\epsilon_\theta$
**Require:** few Steps $K_{\text{few}} = 2$, Full Steps $K_{\text{full}} = 50$
**Require:** Guidance Interval $[k_{\text{start}}, k_{\text{end}})$, Strength $\lambda_0$
 1: *// (1) Motion Prior Extraction*
 2: $\mathbf{z}_T \sim \mathcal{N}(\mathbf{0}, \mathbf{I})$
 3: $\mathbf{z}^{\text{few}} \leftarrow \mathcal{S}(\mathbf{z}_T, \mathbf{c}, K_{\text{few}}; \epsilon_\theta)$,
 4: $\mathbf{M}^{\text{prior}} \leftarrow \mathbf{z}^{\text{few}}_{2:F} - \mathbf{z}^{\text{few}}_{1:F-1}$ {Extract temporal deltas}
 5: *// (2) Guided Generation*
 6: $\mathbf{z} \leftarrow \mathbf{z}_T$ {Re-use initial noise}
 7: **for** $k = 0$ to $K_{\text{full}} - 1$ **do**
 8: $\quad \mathbf{z} \leftarrow \text{Step}(\mathbf{z}, \epsilon_\theta, \mathbf{x}_{ref}, p)$ {One denoising step}
 9: $\quad$ *// Apply Guidance*
10: $\quad$ **if** $k_{\text{start}} \leq k < k_{\text{end}}$ **then**
11: $\quad\quad \mathbf{M} \leftarrow \mathbf{z}_{2:F} - \mathbf{z}_{1:F-1}$ {Current motion}
12: $\quad\quad \mathcal{G} \leftarrow \mathbf{M}^{\text{prior}} - \mathbf{M}$
13: $\quad\quad \lambda \leftarrow \lambda_0 \cdot \left(1 - \frac{k - k_{\text{start}}}{k_{\text{end}} - k_{\text{start}}}\right)$ {Linear decay}
14: $\quad\quad \mathbf{z}_{2:F} \leftarrow \mathbf{z}_{2:F} + \lambda \cdot \mathcal{G}$
15: $\quad$ **end if**
16: **end for**
17: **return** $\text{Decode}(\mathbf{z})$

---

*Table 6.* **Quantitative comparison of Physics-IQ scores across varying inference steps (NFE)**

| Method | NFE (timesteps) | Physics-IQ Score |
|---|---|---|
| Baseline (CogVideoX) | 50 | 30.82 |
| | 51 | 31.44 |
| | 52 | 31.02 |
| | 53 | 31.54 |
| | 54 | 31.14 |
| | 55 | 30.74 |
| | 60 | 31.51 |
| | 80 | 31.94 |
| | 100 | 31.96 |
| + PHASELOCK | 50 (+2) | 36.0 |

## D.3. Why Direct Frequency Manipulation Fails

A natural question arises: if phase information encodes motion dynamics, why not directly inject the low-frequency phase from the 2-step prior into the 50-step generation? In this section, we investigate several direct frequency manipulation baselines and explain why they fail to preserve physical consistency, motivating our spatial-domain Latent Delta Guidance approach.

### D.3.1. BASELINE METHODS

We evaluate four direct frequency manipulation strategies:

**(1) Low-Frequency Phase Injection.** Extract the low-frequency phase from the 2-step prior and inject it into the 50-step latent at each denoising step:

$$\mathcal{F}(\mathbf{z}^{(k)}_{\text{guided}}) = |\mathcal{F}(\mathbf{z}^{(k)})| \cdot \exp\left(j \cdot \left[\mathbf{M}_{\text{LP}} \odot \phi^{\text{prior}} + (1 - \mathbf{M}_{\text{LP}}) \odot \phi^{(k)}\right]\right), \tag{31}$$

where $\mathbf{M}_{\text{LP}}$ is a Gaussian low-pass filter with cutoff frequency $d_s$ (spatial) and $d_t$ (temporal), and $\phi^{\text{prior}}$, $\phi^{(k)}$ denote the phase spectra of the prior and current latent, respectively.

**(2) Full Phase Substitution.** Replace the entire phase spectrum with that of the 2-step prior while retaining the magnitude from the 50-step generation:

$$\mathbf{z}_{\text{guided}}^{(k)} = \mathcal{F}^{-1}\left(|\mathcal{F}(\mathbf{z}^{(k)})| \cdot \exp(j \cdot \phi^{\text{prior}})\right). \tag{32}$$

**(3) Iterative Refinement.** Following FreeInit (Wu et al., 2024), we iteratively refine the initial noise by preserving low-frequency components from the denoised latent and resampling high-frequency components:

$$\mathbf{z}_T^{(i+1)} = \mathcal{F}^{-1}\left(\mathbf{M}_{\text{LP}} \odot \mathcal{F}(\tilde{\mathbf{z}}_T^{(i)}) + (1 - \mathbf{M}_{\text{LP}}) \odot \mathcal{F}(\boldsymbol{\eta})\right), \tag{33}$$

where $\tilde{\mathbf{z}}_T^{(i)}$ is the re-noised latent from iteration $i$ and $\boldsymbol{\eta} \sim \mathcal{N}(0, \mathbf{I})$ is fresh Gaussian noise.

**(4) Magnitude-Preserving Phase Blend.** Linearly blend the phase spectra while preserving the original magnitude:

$$\phi_{\text{blend}} = \alpha \cdot \phi^{\text{prior}} + (1 - \alpha) \cdot \phi^{(k)}, \quad \mathbf{z}_{\text{guided}}^{(k)} = \mathcal{F}^{-1}\left(|\mathcal{F}(\mathbf{z}^{(k)})| \cdot \exp(j \cdot \phi_{\text{blend}})\right). \tag{34}$$

### D.3.2. EXPERIMENTAL SETUP

We conduct experiments on CogVideoX-5B using the Physics-IQ benchmark. For all frequency manipulation methods, we apply guidance at each denoising step $k \in [0, K/2)$ to match the schedule of our proposed method.

Table 7 presents the experimental results comparing direct frequency manipulation methods against our approach.

*Table 7.* **Comparison of frequency manipulation baselines vs. our method on CogVideoX-5B.** Direct frequency manipulation methods substantially underperform the baseline, suggesting that explicit spectral surgery can destabilize the latent representation.

| Method | Physics-IQ ($\uparrow$) | $\Delta$ from Ours |
|---|---|---|
| CogVideoX Baseline (50-step) | 30.82 | -5.06 |
| Magnitude-Preserving Phase Blend ($\alpha$=0.5) | 14.45 | -21.51 |
| Full Phase Substitution | 14.21 | -21.75 |
| Low-Freq Phase Injection | 13.69 | -22.27 |
| Iterative Refinement ($n$=2) | 1.42 | -34.54 |
| **Ours (Latent Delta)** | **35.96** | — |

All frequency manipulation methods substantially underperform even the unmodified baseline. These results suggest that direct spectral manipulation can destabilize the latent representation rather than reliably preserving physical consistency.

### D.3.3. ANALYSIS: WHY DIRECT FREQUENCY MANIPULATION FAILS

We identify three fundamental reasons why direct frequency manipulation fails to preserve physical consistency.

**(1) Spectral Artifacts from FFT Operations.** FFT-based manipulation introduces multiple sources of artifacts. First, the FFT assumes periodic boundary conditions, but video latents have no such periodicity; this mismatch causes *spectral leakage* at spatial and temporal boundaries. Second, frequency-domain filtering produces *ringing artifacts* due to the filter's impulse response, particularly at object boundaries and motion discontinuities where sharp transitions exist. These artifacts propagate through subsequent denoising steps and compound into visible distortions.

**(2) Latent Space Structure Mismatch.** Direct phase manipulation assumes that phase and magnitude form an appropriate decomposition for the learned latent space. However, VAE encoders learn representations optimized for reconstruction fidelity, not spectral separability. We hypothesize that the encoder learns a joint representation where phase and magnitude together encode semantic content in a coupled manner. Substituting phase while preserving magnitude creates hybrid

representations that likely fall outside the learned latent manifold, producing outputs the decoder may not properly reconstruct. This hypothesis is supported by our causal ablation (Sec. 3), which shows that motion is $8.5\times$ more sensitive to phase corruption than magnitude corruption. This strong sensitivity suggests that phase carries critical structural information that cannot be surgically replaced without disrupting the latent's coherence.

**(3) Frequency Domain $\neq$ Physical Property Preservation.** Low-frequency components capture global appearance and coarse motion trajectories, but physical consistency involves constraints that span all frequencies:

- **Conservation laws**: Momentum and energy conservation operate across all spatial frequencies, not just low-frequency bands.

- **Collision dynamics**: Object collisions and contact events involve sharp discontinuities that manifest as high-frequency components.

- **Causal relationships**: Physical causality (cause precedes effect) is a temporal ordering constraint that is not localized to any frequency band.

FreeInit (Wu et al., 2024) improves *temporal smoothness* by preserving low-frequency temporal correlations, but temporal smoothness is neither necessary nor sufficient for physical correctness. A ball smoothly floating upward violates physics despite being temporally consistent, while a ball bouncing with sharp velocity reversals obeys physics despite temporal discontinuities.

### D.3.4. WHY LATENT DELTA GUIDANCE SUCCEEDS

Our Latent Delta Guidance avoids these failure modes through a fundamentally different approach: rather than surgically manipulating spectral components, we constrain the *distribution of inter-frame changes* in the spatial domain.

**(1) Artifact-Free Operation.** By operating entirely in the spatial domain, we never invoke FFT/IFFT operations on the latent representation. This eliminates spectral leakage from boundary discontinuities and ringing artifacts from frequency filtering.

**(2) Aggregate Spectral Constraint via Parseval's Theorem.** While we do not manipulate individual frequency components, Parseval's theorem provides an elegant connection between spatial and frequency domains:

$$\|\mathbf{M}^{\text{prior}} - \mathbf{M}^{(k)}\|^2 \propto \sum_{\omega} \left| \mathcal{F}(\mathbf{M}^{\text{prior}} - \mathbf{M}^{(k)})[\omega] \right|^2. \tag{35}$$

Up to a normalization constant determined by the FFT convention, minimizing the left-hand side in the spatial domain equivalently minimizes the sum of squared spectral differences. Combined with our analysis in §C, which shows that latent delta magnitude is dominated by inter-frame phase differences (under typical video statistics), this provides an *aggregate constraint* on phase evolution without requiring per-frequency intervention.

**(3) Magnitude-Weighted Prioritization.** The aggregate constraint naturally weights frequency components by their squared magnitude $A[\omega]^2$. This is a desirable property: high-magnitude components carry the most perceptually significant motion information, while low-magnitude components (often corresponding to noise or fine texture) contribute minimally to the loss. The guidance thus focuses on dominant motion patterns while allowing flexibility in perceptually less important details.

**(4) Respecting the Denoising Trajectory.** Unlike spectral surgery that forces specific frequency values, our guidance signal $\mathcal{G}^{(k)} = \mathbf{M}^{\text{prior}} - \mathbf{M}^{(k)}$ operates as a soft constraint that nudges the denoising trajectory toward the motion prior. The diffusion model retains the ability to find a coherent solution within its learned manifold, rather than being forced into potentially inconsistent states.

**Key Insight.** The critical difference from direct phase manipulation is that we constrain *how latent representations change between frames*, not their absolute spectral values. This respects the diffusion model's learned dynamics while providing sufficient aggregate guidance to transfer motion priors. The empirical results (Table 7) validate that such aggregate constraints are not only sufficient but substantially more effective than per-frequency surgical intervention—precisely because they avoid disrupting the latent space structure that the model relies upon.

### D.4. Spectral Analysis of PhaseLock

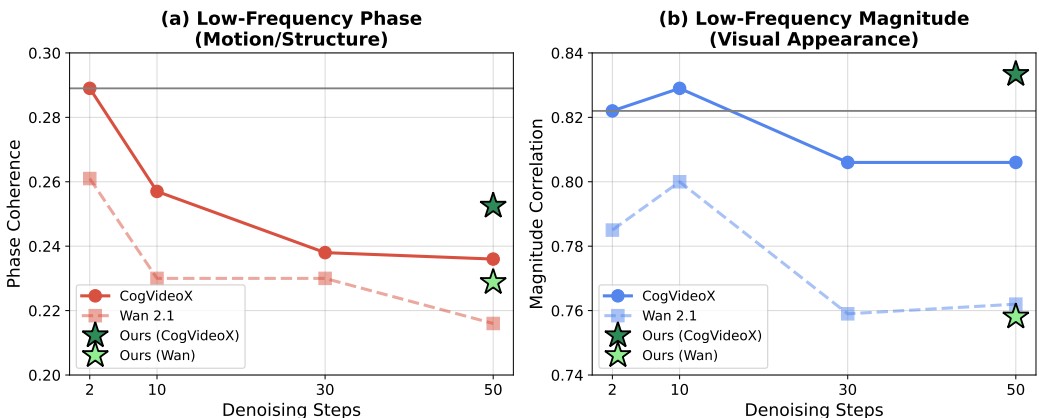

*Figure 9.* **Additional Analysis of PhaseLock.** PhaseLock mitigates phase erosion during the denoising process without explicit FFT operations.

To quantitatively verify that our method preserves phase information as analyzed in Sec. 3, we additionally measured the low-frequency phase coherence and magnitude correlation between the videos generated by our method and the GT videos. We extract low-frequency components (within 40% of the Nyquist frequency) from the 3D FFT of 16-frame sequences, then compute phase coherence as $\mathbb{E}[\cos(\phi_{\text{real}} - \phi_{\text{gen}})]$ and magnitude correlation via Pearson's $r$ on log-magnitude spectra. For baseline models, phase coherence degrades substantially as denoising progresses: CogVideoX drops from 0.289 (2-step) to 0.236 (50-step), an 18.3% relative decrease, while Wan exhibits a similar 17.2% drop (0.261 to 0.216). In contrast, magnitude correlation remains stable across all steps ($<3\%$ variation), confirming that phase, encoding object positions and motion direction, is the vulnerable component during denoising. Notably, our method at 50 steps achieves phase coherence of 0.253 (CogVideoX) and 0.229 (Wan), closely matching the *10-step* baseline levels (0.257 and 0.230) rather than the degraded 50-step values. This suggests that PHASELOCK mitigates phase erosion without explicit FFT operations. By constraining frame deltas in the spatial domain, we indirectly preserve the inter-frame phase differences that encode motion dynamics (Eq. 4), achieving phase preservation while avoiding artifacts from direct spectral manipulation. The corresponding visualization is shown in Fig. 9.

## E. Evaluation Details

To objectively assess physical motion dynamics, we employed the Physics-IQ benchmark (Motamed et al., 2026). Additionally, we utilized PhyGenBench (Meng et al., 2025), which leverages Vision-Language Models (VLMs) for evaluation, as well as VBench (Huang et al., 2024). To mitigate the potential limitations of automated metrics and ensure a robust evaluation, we complemented these methods with a Human Study, strictly adhering to the protocols established in WMReward (Yuan et al., 2026). A detailed description of each evaluation methodology is provided below.

**Physics-IQ Benchmark.** To evaluate the physical understanding of video generation models, we utilized the Physics-IQ benchmark (Motamed et al., 2026). This benchmark comprises a diverse collection of 396 high-quality, real-world videos covering 66 distinct physical scenarios, ranging from fundamental principles such as solid mechanics and fluid dynamics to complex phenomena like optics and magnetism. The evaluation protocol requires models to predict a 5 second video continuation based on initial conditioning frames. Performance is quantitatively assessed by comparing the generated motion against Ground Truth (GT) videos using a set of physics-aware metrics (including Spatial IoU, Spatiotemporal IoU, and Mean Squared Error (MSE)), which are aggregated into a unified Physics-IQ score to measure the model's adherence to real-world physical laws.

**PhyGenBench Evaluation.**    We also incorporated PhyGenBench (Meng et al., 2025) to assess the model's grasp of physical commonsense through a text-to-video generation task. This benchmark consists of 160 carefully crafted prompts designed to probe 27 distinct physical laws across four fundamental domains: mechanics, optics, thermal dynamics, and material properties. Unlike reference-based metrics, PhyGenBench employs PhyGenEval, a hierarchical evaluation framework that leverages advanced Vision-Language Models (VLMs) and Large Language Models (LLMs) to automate the assessment process. The evaluation is conducted in three progressive stages: Key Physical Phenomena Detection, which verifies the presence of specific physical events; Physics Order Verification, which checks the causal and temporal sequence of these events; and Overall Naturalness Evaluation, which rates the global realism of the video. This multi-stage approach ensures that the generated content is evaluated not just for visual quality, but for its adherence to fundamental physical principles.

Since PhyGenBench is originally designed for text-to-video (T2V) tasks, adapting it to our framework required input images to serve as initial conditions. To address this, we synthesized reference images using two distinct text-to-image (T2I) models: the 'gemini-2.5-flash-image-preview' (Comanici et al., 2025) and 'FLUX-schnell' (Esser et al., 2024) as shown in Fig. 10. In generating these inputs, we modified the original PhyGenBench prompts to explicitly depict the scene immediately preceding the onset of motion (*e.g.,* Original Prompt + ", static scene immediately before the action."). This temporal positioning facilitates a more precise comparison of the generated motion's intensity and velocity. In the main paper, we provide a comparative analysis of results based on both image sources to test generalizability. Furthermore, we acknowledge an inevitable domain gap between these synthetic starting frames and real-world imagery; thus, the evaluations reflect performance within a fully synthetic generation pipeline.

### Images Generated by Gemini-2.5-flash

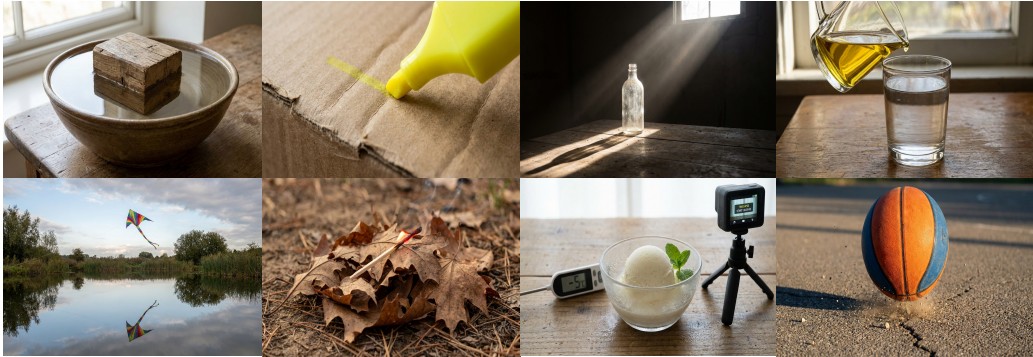

### Images Generated by FLUX-schnell

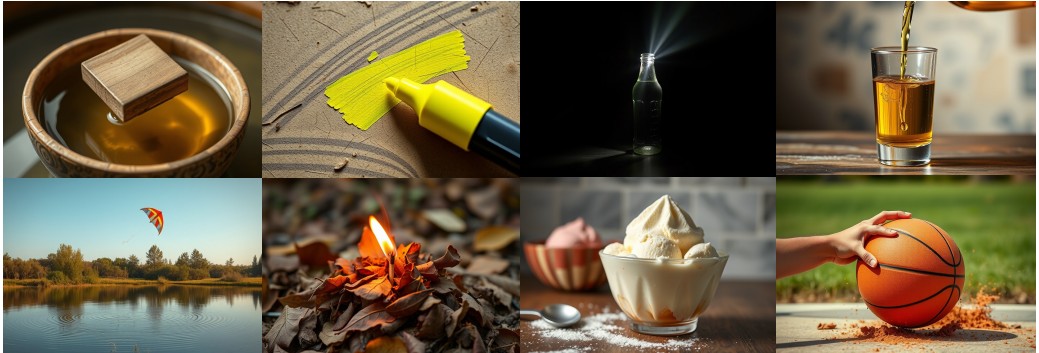

*Figure 10.* **Sample images from PhyGenBench generated by Gemini-2.5 Flash and FLUX-schnell, serving as input frames for the Image-to-Video (I2V) task.**

**VBench Evaluation.**    Following the comprehensive evaluation protocol of (Yuan et al., 2026), we employed Physics-IQ and PhyGenBench to assess physical fidelity, alongside VBench (Huang et al., 2024) for general video quality. While our primary focus is on physical consistency, it is essential to ensure that the proposed method maintains high standards in fundamental generation metrics. Therefore, we utilized VBench to specifically evaluate key dimensions including Image Quality, Aesthetic Quality, Motion Smoothness, and Temporal Consistency. This verification ensures that the improvements

*Table 8.* **Human Evaluation Results.** We compare the Win Rate and Accuracy of our method against the baseline (CogVideoX-5B) across three categories. Note that Accuracy accounts for ties with a weight of 0.5.

| | Win Rate (%) | | Accuracy (%) | |
|---|---|---|---|---|
| **Category** | **Ours** | **Baseline** | **Ours** | **Baseline** |
| Physics Plausibility | **78.3** | 21.7 | **58.5** | 41.5 |
| Visual Quality | **78.9** | 21.1 | **61.7** | 38.3 |
| Prompt Alignment | **60.4** | 39.6 | **54.0** | 46.0 |

*Table 9.* **Human Evaluation Results.** We compare the Win Rate and Accuracy of our method against the baseline (Wan 2.1) across three categories. Note that Accuracy accounts for ties with a weight of 0.5.

| | Win Rate (%) | | Accuracy (%) | |
|---|---|---|---|---|
| **Category** | **Ours** | **Baseline** | **Ours** | **Baseline** |
| Physics Plausibility | **83.3** | 16.7 | **70.1** | 29.9 |
| Visual Quality | **88.2** | 11.8 | **74.7** | 25.3 |
| Prompt Alignment | **78.5** | 21.5 | **65.8** | 34.2 |

in physical alignment do not compromise the overall visual fidelity and temporal coherence of the videos.

**Human Study Protocol.** To complement our quantitative metrics, we conducted a human preference study strictly following the protocol of (Yuan et al., 2026). We presented annotators with pairwise video comparisons (Ours vs. Baseline) in a randomized, blind manner to eliminate potential bias. The evaluation was conducted across three distinct criteria:

- *Physics Plausibility:* Which video better follows realistic physical dynamics?

- *Prompt Alignment:* Which video better corresponds to the input text prompt?

- *Visual Quality:* Which video exhibits higher visual fidelity?

For each criterion, annotators were asked to select the superior video or indicate a tie if both were of similar quality. We report two metrics: Win Rate and Accuracy. Win Rate excludes ties and is computed as:

$$\text{Win Rate} = \frac{N_{\text{win}}}{N_{\text{win}} + N_{\text{loss}}},$$

where $N_{\text{win}}$ and $N_{\text{loss}}$ denote the number of non-tied comparisons won and lost by the corresponding method, respectively.

To include neutral results in the evaluation rather than discarding them, we assign a score of $0.5$ to each tie. Accuracy is calculated as:

$$\text{Accuracy} = \frac{N_{\text{win}} + 0.5 \times N_{\text{tie}}}{N_{\text{total}}},$$

where $N_{\text{tie}}$ and $N_{\text{total}}$ denote the number of ties and total comparisons, respectively.

The overall results are shown below in Table 8, and the corresponding Wan 2.1 breakdown is provided in Table 9.

# F. Additional Experiment Results

## F.1. Additional Ablation Studies

In this section, we present additional ablation studies that were not included in the main paper due to space constraints. Specifically, we evaluate the effectiveness of our adaptive scheduling by comparing it with alternative scheduling strategies. We also analyze different guidance formulations to validate our choice of using Latent Delta guidance. Finally, we investigate the impact of the timestep range when applying this guidance. All experiments are conducted on CogVideoX-5B using the Physics-IQ benchmark.

### F.1.1. SCHEDULING ABLATIONS

We compare our linear decay schedule against four alternative scheduling strategies, all using the same initial guidance strength ($\lambda_0 = 0.05$) and active range.

Table 10 presents the results.

*Table 10.* **Ablation on guidance scheduling strategies.** Linear decay achieves the best balance between early-stage motion anchoring and late-stage refinement flexibility.

| Schedule | Formulation | Physics-IQ |
|---|---|---|
| Exponential | $\lambda_0 \cdot e^{-\alpha k}$ | 34.9 |
| Cosine | $\lambda_0 \cdot \frac{1+\cos(\pi\rho)}{2}$ | 32.9 |
| Constant | $\lambda_0$ | 31.3 |
| Step | $\lambda_0 \cdot \mathbf{1}_{[k<k_{\text{end}}]}$ | 31.3 |
| Linear (Ours) | $\lambda_0 \cdot \left(1 - \frac{k-k_{\text{start}}}{k_{\text{end}}-k_{\text{start}}}\right)$ | **36.0** |

The results reveal that gradual decay is essential for optimal performance. Constant scheduling (31.32) applies uniform guidance throughout, which over-constrains the model during later timesteps when high-frequency visual details should be refined freely. Step scheduling shows identical performance, confirming that abrupt termination provides no benefit over constant guidance within the active range.

Exponential decay (34.91) improves over constant by reducing guidance more rapidly, but its aggressive early decay ($e^{-\alpha k}$ drops to $< 10\%$ of $\lambda_0$ by mid-range) releases the motion constraint too quickly, allowing phase drift to accumulate. Cosine scheduling (32.92) suffers from the opposite problem: it maintains near-maximum guidance for too long before rapidly dropping, creating a discontinuity in the guidance profile.

Our linear decay achieves the best performance (36.0) by providing strong guidance during early timesteps when coarse motion structure is established (where phase preservation is most critical per Sec. 3), while smoothly releasing the constraint to allow the model to refine visual details. This matches the intuition from our phase erosion analysis: motion dynamics encoded in low-frequency phase are most vulnerable during early-to-mid denoising, and the guidance should taper proportionally as the generation progresses toward high-frequency refinement.

### F.1.2. GUIDANCE FORMULATION ABLATION

We compare our Latent Delta guidance against three alternative formulations for transferring motion information from the prior. Table 11 shows the results.

*Table 11.* **Ablation on guidance formulations.** Latent Delta guidance on frame differences significantly outperforms alternatives.

| Formulation | Definition | Physics-IQ |
|---|---|---|
| Normalized | $\frac{\mathbf{M}^{\text{prior}}-\mathbf{M}^{(k)}}{\|\mathbf{M}^{\text{prior}}\|_2}$ | 31.3 |
| Direct Latent | $\mathbf{z}^{\text{prior}} - \mathbf{z}^{(k)}$ | 16.0 |
| Second Order | $\mathbf{A}^{\text{prior}} - \mathbf{A}^{(k)}$ | 11.9 |
| Latent Delta (Ours) | $\mathbf{M}^{\text{prior}} - \mathbf{M}^{(k)}$ | **36.0** |

- **Direct Latent** guidance ($\mathbf{z}^{\text{prior}} - \mathbf{z}^{(k)}$) directly constrains the latent toward the 2-step output. This performs poorly (15.97) because it conflates static content with motion dynamics: the guidance fights against legitimate visual refinement that improves appearance while inadvertently degrading physics. This suggests that *what* is transferred matters as much as *how*.

- **Second Order** guidance uses frame accelerations $\mathbf{A} = \mathbf{M}_{2:F} - \mathbf{M}_{1:F-1}$ (second temporal derivative), motivated by the intuition that accelerations directly encode physical forces. However, this performs worst (11.89) because second-order differences amplify high-frequency noise present in the 2-step prior, which has lower visual fidelity. The guidance becomes dominated by noise rather than meaningful dynamics.

- **Normalized** guidance scales the delta by prior magnitude, intended to provide scale-invariant steering. This under-performs (31.34) because normalization disrupts the natural relationship between guidance magnitude and motion amplitude, large motions require proportionally larger corrections, which normalization prevents.

Our Latent Delta guidance operates on first-order frame differences without normalization, directly targeting the velocity field that encodes motion dynamics. Per our theoretical analysis (Sec. 4), frame deltas $\mathbf{M} = \mathbf{z}_{2:F} - \mathbf{z}_{1:F-1}$ encode motion phase information while being invariant to static scene content, achieving precise motion transfer without interfering with appearance refinement.

### F.1.3. HYPERPARAMETER SENSITIVITY.

We provide extended analysis of the three key hyperparameters introduced in Sec. 5: guidance strength $\lambda_0$, number of few-step inference steps ($K_{\text{fast}}$, NFE), and guidance end step $k_{\text{end}}$. While the main paper presents the overall trends, here we discuss the underlying mechanisms and practical implications in detail.

**Guidance Strength ($\lambda_0$).** As shown in the main paper, performance exhibits a clear inverted-U pattern with peak at $\lambda_0 = 0.05$ (36.0). The mechanisms behind this are twofold. For weak guidance ($\lambda_0 = 0.03$, 34.1), the steering signal is insufficient to counteract the model's tendency toward phase drift during denoising, the latent trajectory deviates from the motion prior before correction can accumulate. For strong guidance ($\lambda_0 \geq 0.10$), the generation becomes over-constrained: the model cannot refine high-frequency visual details because guidance forces it to remain too close to the low-fidelity 2-step prior. At $\lambda_0 = 0.13$, performance degrades to 31.3, which is comparable to or slightly below the CogVideoX baseline (31.32). This indicates that guidance strength must balance two competing objectives: (1) anchoring the motion trajectory to preserve phase information, and (2) allowing sufficient freedom for the model to refine textures and details. The optimal $\lambda_0 = 0.05$ achieves this balance, providing enough correction to preserve phase evolution while permitting visual refinement orthogonal to motion.

**Number of Few-Step Inference Steps (NFE).** Performance monotonically decreases as NFE increases, and the trend is consistent across backbones. On CogVideoX, the Physics-IQ score drops from 36.0 at NFE=2 to 32.8 at NFE=5 and 30.5 at NFE=10. On Wan 2.1, we observe the same monotonic pattern with a gentler slope: $28.7 \rightarrow 28.4 \rightarrow 27.9 \rightarrow 26.2 \rightarrow 24.2$ as NFE increases from 2 through 10. This result is consistent with our explanation that phase erosion accumulates during denoising. Each additional step in the prior inference can introduce more phase corruption, degrading the quality of the motion signal before it is even used for guidance. The steep initial drop from NFE=2 to NFE=5 (CogVideoX: $36.0 \rightarrow 32.8$) is particularly notable, suggesting that phase corruption is most severe in early-to-mid denoising iterations where coarse motion structure is established. Beyond NFE=5, degradation continues but at a slower rate, suggesting that the majority of phase damage occurs early. The consistency of this monotonic degradation across architecturally distinct backbones (DiT and Expert Transformer) supports the view that phase erosion is not a quirk of any one model. Practically, this means our method achieves optimal results with the fewest possible prior steps (NFE=2), which also minimizes computational overhead.

**Guidance End Step ($k_{\text{end}}$).** Fixing $k_{\text{start}} = 0$, we vary $k_{\text{end}}$ from 15 to 40 with $K_{\text{full}} = 50$. Performance peaks at $k_{\text{end}} = 25$ (36.0), corresponding to guidance during exactly the first half of denoising. The results reveal an interesting asymmetry: terminating guidance too early ($k_{\text{end}} = 15$, 35.8; $k_{\text{end}} = 20$, 35.0) causes mild degradation as motion structure is not yet fully established when guidance stops. However, extending guidance too long shows even milder effects ($k_{\text{end}} = 30$, 35.4; $k_{\text{end}} = 40$, 35.3), indicating that late-stage guidance neither helps nor significantly hurts; by this point, the coarse motion trajectory is already largely established. The relatively flat profile across $k_{\text{end}} \in [15, 40]$ (all scores within 35.0–36.0) demonstrates that our method is robust to this hyperparameter, simplifying practical deployment. We recommend $k_{\text{end}} = K_{\text{full}}/2$ as a principled default that balances motion anchoring with refinement freedom.

## F.2. Architectural Generality

To further verify that phase erosion is not an artifact of a specific backbone, we evaluate PhaseLock on three additional architectures spanning distinct design paradigms: SVD (UNet-based), SkyReels-V2 (different scheduler and resolution), and Wan 2.1 Distilled (Lightweight DiT, 4-step). As reported in Table 12, PhaseLock yields consistent improvements across all architectural families, with a particularly large absolute gain on SkyReels-V2 (+8.9). The smaller gain on SVD (+1.2) is consistent with the structural property of UNet-based models, which exhibit less inter-frame phase coupling in latent space compared to Transformer-based backbones, and should therefore be read as supporting evidence for our phase-erosion

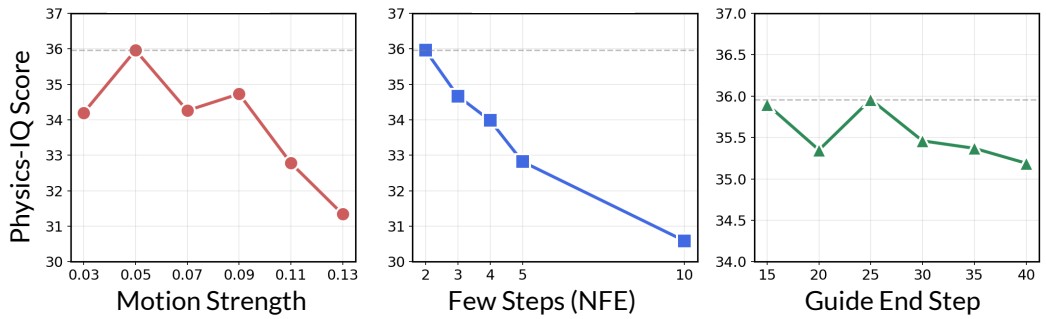

*Figure 11.* **Ablation Studies of PhaseLock**

analysis rather than a limitation of PhaseLock.

*Table 12.* **Architectural generality on Physics-IQ.** PhaseLock yields consistent gains across UNet, Expert Transformer, Lightweight DiT, and standard DiT backbones.

| Architecture | Model | Score | Gain |
|---|---|---|---|
| UNet | SVD (Base) | 14.82 | – |
| | **+ PhaseLock** | 15.98 | **(+1.2)** |
| Different Scheduler | SkyReels-V2 (Base) | 24.68 | – |
| | **+ PhaseLock** | 33.61 | **(+8.9)** |
| Lightweight DiT | Wan 2.1 Distilled (LightX2V, 4-step) | 27.7 | – |
| | **+ PhaseLock** | 29.4 | **(+1.7)** |

### F.3. Per-Scenario Performance Characterization

The main-text Physics-IQ score (Table 1) reports an average across 66 scenarios, which can hide large heterogeneity in how PhaseLock behaves on different types of dynamics. Here we break the score down along two axes: rigid vs. non-rigid motion, and per-category Physics-IQ scenarios for each backbone.

**Scale and Bounded Degradation.** Across all 66 Physics-IQ scenarios, PhaseLock improves motion dynamics on 74% (49/66) of scenarios for Wan 2.1 and 67% (44/66) for CogVideoX-5b. The worst per-scenario degradation is $-20.3$ on Wan 2.1 and $-22.2$ on CogVideoX-5b, which is less than half of the best per-scenario improvement ($+55.4$ and $+57.9$, respectively). The impact of the guidance is therefore bounded when it hurts and substantial when it helps, as shown in Fig. 14 and Table 13.

**Rigid vs. Non-Rigid Dynamics.** Of the 66 Physics-IQ scenarios, 27 (41%) involve non-rigid dynamics: 15 fluid scenarios (e.g., pouring, capillary flow, siphon), 9 deformable solids (e.g., cloth draping, soft-body cutting, granular deformation), and 3 thermodynamic scenarios (e.g., combustion, balloon rupture). On average, PhaseLock improves non-rigid scenarios by $+41.8\%$, compared with $+23.4\%$ on rigid-body scenarios. This is consistent with phase sensitivity being most visible where motion is continuous, directional, and velocity-dominated rather than impulsive; see Table 17. Representative non-rigid examples are shown in Fig. 12, while scenarios that violate or obey physical laws are illustrated in Fig. 13.

**Representative Scenarios.** The largest per-scenario gains concentrate in complex deformable and fluid dynamics: napkin-soak ($+54.7$, capillary flow), siphon ($+55.4$, fluid transfer), cut-orange ($+51.9$, rapid deformation with sharp boundary), silk-cover ($+35.8$, soft-body drape on CogVideoX), and potato-in-water ($+18.3$, fluid displacement). These scenarios are typical failure cases for standard 50-step generation, which tends to either stall motion or produce implausible acceleration.

**High-Frequency and Instantaneous Dynamics.** A potential concern raised is whether locking low-frequency phase (the regime we use for diagnosis in Sec. 3) is sufficient to capture sharp-boundary, high-frequency events such as fracture, shattering, or combustion. We emphasize that Latent Delta Guidance itself operates over the full spatial domain rather than a single frequency band: by Parseval's theorem, the L2 constraint on the delta aggregates across all frequencies. Empirically,

**Soft-body Floating:** A wine cork is dropped into a full glass of red wine, splashing lightly and then floating on the surface.

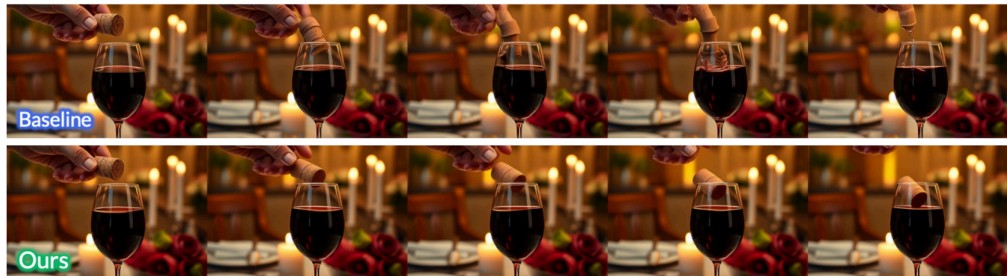

**Cloth Folding:** A soft red cloth slowly slides off the edge of a wooden table and folds onto the floor.

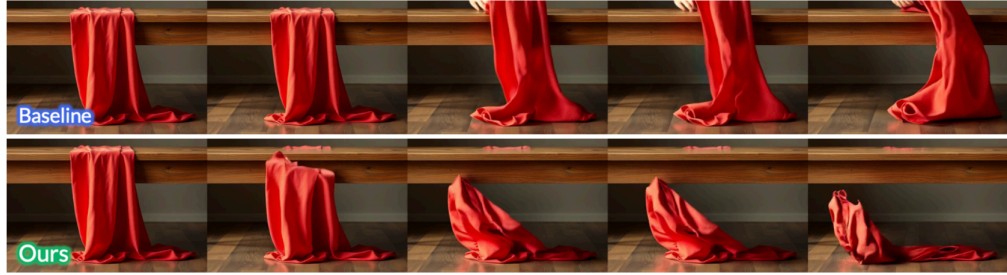

*Figure 12.* **Examples of specific non-rigid physics scenarios.**

*Table 13.* **Overall impact of our method on Physics-IQ scores across two base models.**

| Metric | Wan 2.1 | CogVideoX-5b |
|---|---|---|
| Overall $\Delta$ (official) | +7.8 | +5.2 |
| Improved | 49/66 | 44/66 |
| Degraded | 13/66 | 18/66 |
| Unchanged | 4/66 | 4/66 |
| Max improved | +55.4 | +57.9 |
| Max degraded | −20.3 | −22.2 |

scenarios involving sharp transitions (cut-orange +51.9, siphon +55.4, match-blows-balloon) are among the largest-gain cases, confirming that the adaptive linear decay of $\lambda$ leaves later denoising steps free to render high-frequency detail while the early steps establish the inter-frame velocity field. The corresponding ablation curves are summarized in Fig. 11.

**Per-Category Breakdown.** The per-category Physics-IQ results in Tables 14 and 15 show consistent improvements across the major physical categories, with the largest relative gains appearing in categories most closely tied to non-rigid motion. This aligns with our rigid vs. non-rigid observation above: non-rigid material response is precisely the regime where phase preservation matters most. The finer-grained category summary and rigid/non-rigid comparison are reported in Tables 16 and 17.

To further localize where the gains arise, we summarize the improvement by specific physical category and by rigid/non-rigid scenario split in the tables below.

### F.4. More Qualitative Results

In this section, we present additional qualitative results on Physics-IQ in Fig. 19–Fig. 22 and on PhyGenBench in Fig. 23-Fig. 26.

*Note.* Full-page figures are placed at the bottom of the document.

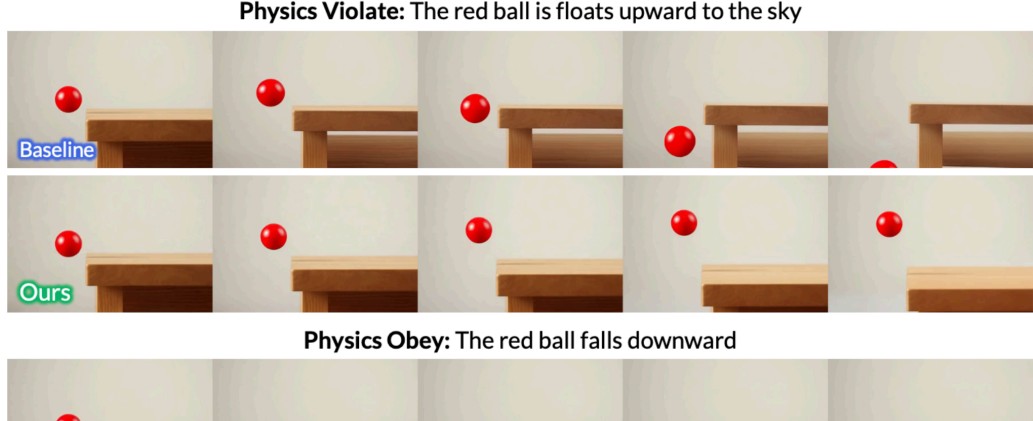

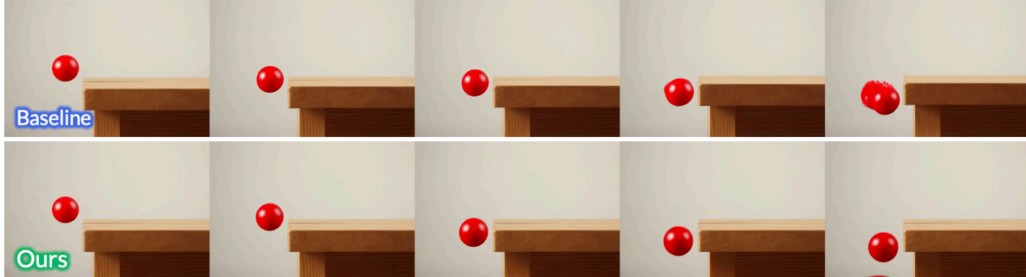

*Figure 13.* **Examples of scenarios that violate or obey the laws of physics.**

*Table 14.* Per-category breakdown for WAN 2.1.

| Category | N | Improved | Degraded | Avg $\Delta$ |
|---|---|---|---|---|
| Fluid Dynamics | 15 | 14 (93%) | 1 (7%) | +12.8 |
| Solid Mechanics | 38 | 27 (71%) | 10 (26%) | +6.8 |
| Thermodynamics | 3 | 2 (67%) | 0 (0%) | +13.2 |
| Optics | 8 | 5 (62%) | 1 (12%) | +1.7 |
| Magnetism | 2 | 1 (50%) | 1 (50%) | +6.8 |

# G. Limitations and Further Discussions

## G.1. Limitations and Future Works

While PhaseLock demonstrates significant improvements in physical consistency, we acknowledge several limitations and failure cases inherent to our approach.

**Dependence on Few-Step Inference Quality.** Our method relies on the 2-step generation to provide a physically plausible motion prior. When the few-step inference itself produces incorrect physics, for instance, due to ambiguous input images or prompts that contradict physical intuition, the guidance will steer the high-fidelity generation toward this erroneous trajectory (Fig. 15). Furthermore, because PhaseLock does not inject external physical knowledge, it may still produce erroneous outputs when the flawed 2-step prior originates from the model's intrinsic limitations (Fig. 15). However, our guidance strength $\lambda$ is moderate and does not force exact replication, allowing the model some flexibility to deviate when strong textural cues conflict with the prior.

**Architecture Specificity.** PhaseLock is designed for diffusion-based video generation, where the iterative denoising process allows us to inject guidance at each step. The method is not directly applicable to autoregressive models such as VideoPoet (Kondratyuk et al., 2024) and MAGI-1 (Teng et al., 2025). These generate frames sequentially without a denoising loop, so there is no opportunity to inject inter-frame guidance.

**Trade-off with Creative Generation.** By guiding motion toward the few-step prior, PhaseLock may constrain artistic or physically unrealistic motions that users intentionally desire. This can be addressed by adjusting $\lambda_0$ or disabling guidance entirely for such use cases.

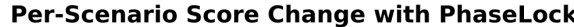

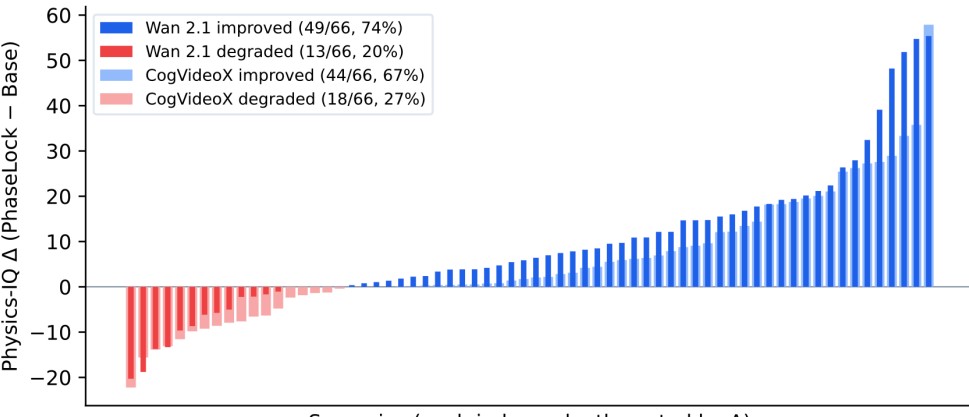

*Figure 14.* Distribution of per-scenario Physics-IQ score changes ($\Delta$) after applying our method. Blue and red bars indicate improved and degraded scenarios, respectively.

*Table 15.* Per-category breakdown for CogVideoX-5B.

| Category | N | Improved | Degraded | Avg $\Delta$ |
|---|---|---|---|---|
| Optics | 8 | 7 (88%) | 1 (12%) | +7.1 |
| Solid Mechanics | 38 | 28 (74%) | 7 (18%) | +6.6 |
| Magnetism | 2 | 1 (50%) | 1 (50%) | +2.7 |
| Fluid Dynamics | 15 | 7 (47%) | 8 (53%) | +2.0 |
| Thermodynamics | 3 | 1 (33%) | 1 (33%) | +0.8 |

## G.2. Further Discussions

In this section, we discuss the broader implications of our findings and outline potential extensions of PhaseLock.

For a broader comparison with other physics-guided video generation methods, Table 18 summarizes the trade-offs in performance, cost, and applicability.

### G.2.1. WHY DOES PHASE EROSION OCCUR?

Our analysis identifies *what* happens (phase degrades more than magnitude) but does not fully explain *why* the denoising process exhibits this asymmetry. We hypothesize several contributing factors:

**Training Objective Bias.** Diffusion models are trained with MSE loss in pixel/latent space, which decomposes into magnitude and phase components in the frequency domain. By Parseval's theorem:

$$\|\mathbf{x} - \hat{\mathbf{x}}\|_2^2 = \|\mathcal{F}(\mathbf{x}) - \mathcal{F}(\hat{\mathbf{x}})\|_2^2. \tag{36}$$

However, the gradient contribution from phase errors is modulated by magnitude:

$$\frac{\partial \text{MSE}}{\partial \phi} \propto A \cdot \sin(\phi - \hat{\phi}). \tag{37}$$

In high-frequency regions where $A$ is small (fine details), phase gradients vanish, causing the model to prioritize magnitude alignment over phase alignment in these regions. Since motion dynamics often involve subtle positional shifts (small $A$, significant $\Delta\phi$), the training objective may inherently under-weight physical consistency.

**Perceptual Loss Landscape.** Human perception is more sensitive to texture (magnitude) than to small positional errors (phase). Training data curation and loss functions optimized for perceptual quality may implicitly encourage magnitude preservation at the expense of phase fidelity.

*Table 16.* **Performance gain by specific physical categories.**

| Category | CogVideoX Gain | WAN 2.1 Gain |
|---|---|---|
| Thermal (phase transitions, combustion) | +11.9% | +28.9% |
| Material (dissolution, deformation, chemical change) | +14.0% | +36.7% |

*Table 17.* **Comparison of improvement between rigid-only and non-rigid scenarios.**

| Model | Rigid-only (39) | Non-rigid (27) | All (66) |
|---|---|---|---|
| CogVideoX | $34.2 \rightarrow 40.7$ (+18.8%) | $42.9 \rightarrow 47.3$ (+10.1%) | $37.8 \rightarrow 43.4$ (+14.7%) |
| WAN 2.1 | $25.8 \rightarrow 31.8$ (+23.4%) | $28.1 \rightarrow 39.8$ (+41.8%) | $26.7 \rightarrow 35.1$ (+31.3%) |

**Coarse-to-Fine Generation Dynamics.** Diffusion models generate global structure before local details (Choi et al., 2022). Phase encodes structure ("where things are"), so it is established early and becomes a target for subsequent refinement. If the model's capacity is allocated toward high-frequency detail generation in later steps, phase information may be treated as a "fixed" quantity to be overwritten rather than preserved.

**Future work**: A rigorous theoretical analysis of phase dynamics under different training objectives (MSE, perceptual loss, adversarial loss) could inform the design of physics-aware training procedures.

### G.2.2. TOWARD PHYSICS-AWARE SAMPLERS

Our finding that phase erosion accumulates across denoising steps suggests that the **sampling algorithm itself** could be modified to minimize phase corruption.

**Phase-Preserving Noise Schedules.** Standard noise schedules (linear, cosine, EDM (Karras et al., 2022)) are designed to balance signal-to-noise ratio for visual quality. A physics-aware schedule could:

- Allocate more denoising capacity to early steps where phase is being established.

- Reduce the number of steps in the mid-to-late range where phase erosion is most severe.

- Introduce phase-weighted loss terms during sampling (though this would require gradient computation).

**Frequency-Adaptive Sampling.** Rather than applying uniform denoising across all frequency bands, an adaptive sampler could:

- Denoise low frequencies (magnitude-dominant) with standard schedules.

- Apply reduced noise perturbation to phase-dominant components.

- Implement band-specific step counts based on convergence.

This would require decomposing the latent into frequency bands at each step, which introduces computational overhead but could yield physics improvements without external guidance.

### G.2.3. EXTENSION TO OTHER MODALITIES

While we focus on video generation, the phase erosion phenomenon may generalize to other sequential generation tasks.

**(1) Audio Generation:** In audio diffusion models (e.g., AudioLDM (Liu et al., 2023)), phase encodes temporal alignment and pitch, while magnitude encodes timbre and loudness. If similar phase erosion occurs, it could manifest as temporal misalignment (sounds occurring at wrong times), Pitch drift, and Loss of rhythmic structure. A PhaseLock-inspired audio method could extract rhythm/pitch priors from few-step inference and guide subsequent generation.

**(2) 3D Generation:** Extracting structural priors from few-step 3D inference could improve geometric plausibility.

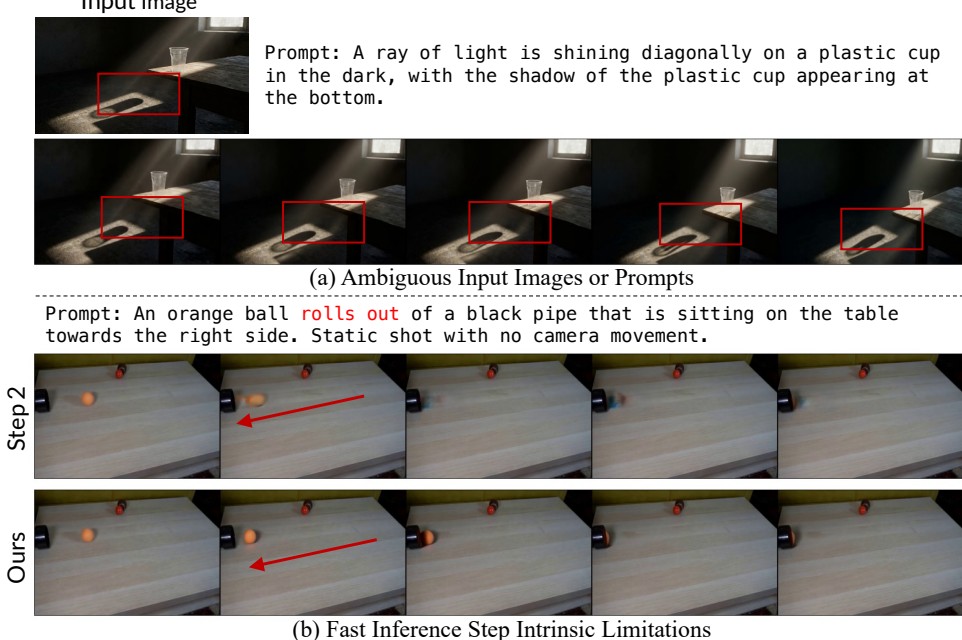

(a) Ambiguous Input Images or Prompts

(b) Fast Inference Step Intrinsic Limitations

*Figure 15.* **Failure Cases of Our Method**

*Table 18.* **Comparison of our method with other physics-guided video generation methods. * Not directly applicable to the full Physics-IQ I2V setting without scenario-specific manual specification.**

| Method | Type | Physics-IQ | Time | Memory | Ext. Model / I2V |
|---|---|---|---|---|---|
| PhaseLock | Inference guidance | 36.0 | $\times 1.06$ | $\times 1.02$ | None / ✓ |
| WMReward($\nabla$ + BoN) | Reward optimization | 36.3 | $\times 5.02N$ | $\times 4.27$ | VJEPA-2 / ✓ |
| PhysGen | External simulation | – | – | – | Rigid-body sim / $\times^*$ |
| PhysCtrl | External simulation | – | – | – | 6-stage pipeline / $\times^*$ |
| VideoREPA | Training alignment | – | – | – | DINO / $\times$ (T2V only) |
| WISA | Training data | – | – | – | Curated data / $\times$ (T2V only) |

**(3) Multimodal Generation.** For joint text-image-video generation, phase erosion could cause cross-modal misalignment (*e.g.,* narration describing action that doesn't match the visual motion). Locking temporal dynamics across modalities could improve coherence.

G.2.4. THEORETICAL EXTENSIONS

**Information-Theoretic Analysis.** Our empirical observation that phase degrades $\sim$18% from step 2 to step 50 while magnitude remains stable suggests an information-theoretic asymmetry. Future work could formalize this through mutual information between phase/magnitude and physical consistency, rate-distortion analysis of phase versus magnitude under denoising, or an information bottleneck perspective on what is preserved versus lost.

**Optimal Transport Perspective.** Diffusion can be viewed as optimal transport from noise to data distribution. Phase erosion suggests the transport path is "curved" in a way that preserves magnitude but corrupts phase. Analyzing the Wasserstein geometry of this transport could reveal why phase is more vulnerable and how to design straighter (phase-preserving) paths.

**Connection to Memorization vs. Generalization.** Phase encodes specific structural details ("where exactly is the ball?"), while magnitude encodes general appearance ("what does a ball look like?"). The phase erosion phenomenon may relate to the model's tendency to generalize appearance while forgetting specific configurations, a form of "structural forgetting" during generation. This connects to broader questions about what diffusion models memorize vs. generalize.

## G.2.5. PRACTICAL EXTENSIONS

**User-Controllable Physics.** Currently, PhaseLock extracts the motion prior automatically from few-step inference. Extensions could allow users to specify desired motion trajectories (e.g., "ball falls at 45° angle"), interpolate between multiple motion priors for controllable dynamics, and apply physics constraints from external simulators as soft priors.

**Long-Video Generation.** For videos longer than the model's native context, PhaseLock could be extended to extract motion priors for each temporal chunk, ensure consistency across chunk boundaries via overlapping guidance, and implement hierarchical priors at multiple temporal scales.

**Real-Time Applications.** The $1.06\times$ overhead of PhaseLock is already low, but for real-time applications (interactive generation, streaming), further optimization could include caching the 2-step prior across frames for video continuation, amortizing prior extraction across multiple outputs, and distilling the guidance into the model weights (one-time training cost).

## G.2.6. SOCIETAL CONSIDERATIONS

**Beneficial Applications.** Physically consistent video generation enables:

- **Robotics**: Training policies on realistic simulated environments

- **Scientific visualization**: Accurate depiction of physical phenomena for education

- **Autonomous vehicles**: Generating diverse, physically plausible driving scenarios for testing

- **Accessibility**: Creating realistic visual descriptions for visually impaired users

**Potential Misuse.** Improved physical realism could make synthetic media harder to distinguish from real footage, potentially enabling:

- More convincing deepfakes

- Fabricated evidence of events that didn't occur

- Misinformation that exploits physical plausibility as a trust signal

**Mitigation**: We advocate for:

- Developing detection methods that identify PhaseLock-specific artifacts

- Watermarking generated content at the latent level

- Responsible disclosure practices that balance openness with harm reduction

Our contribution to understanding *how* physical consistency is achieved (phase preservation) also contributes to understanding *how* to detect synthetic content (by analyzing phase characteristics).

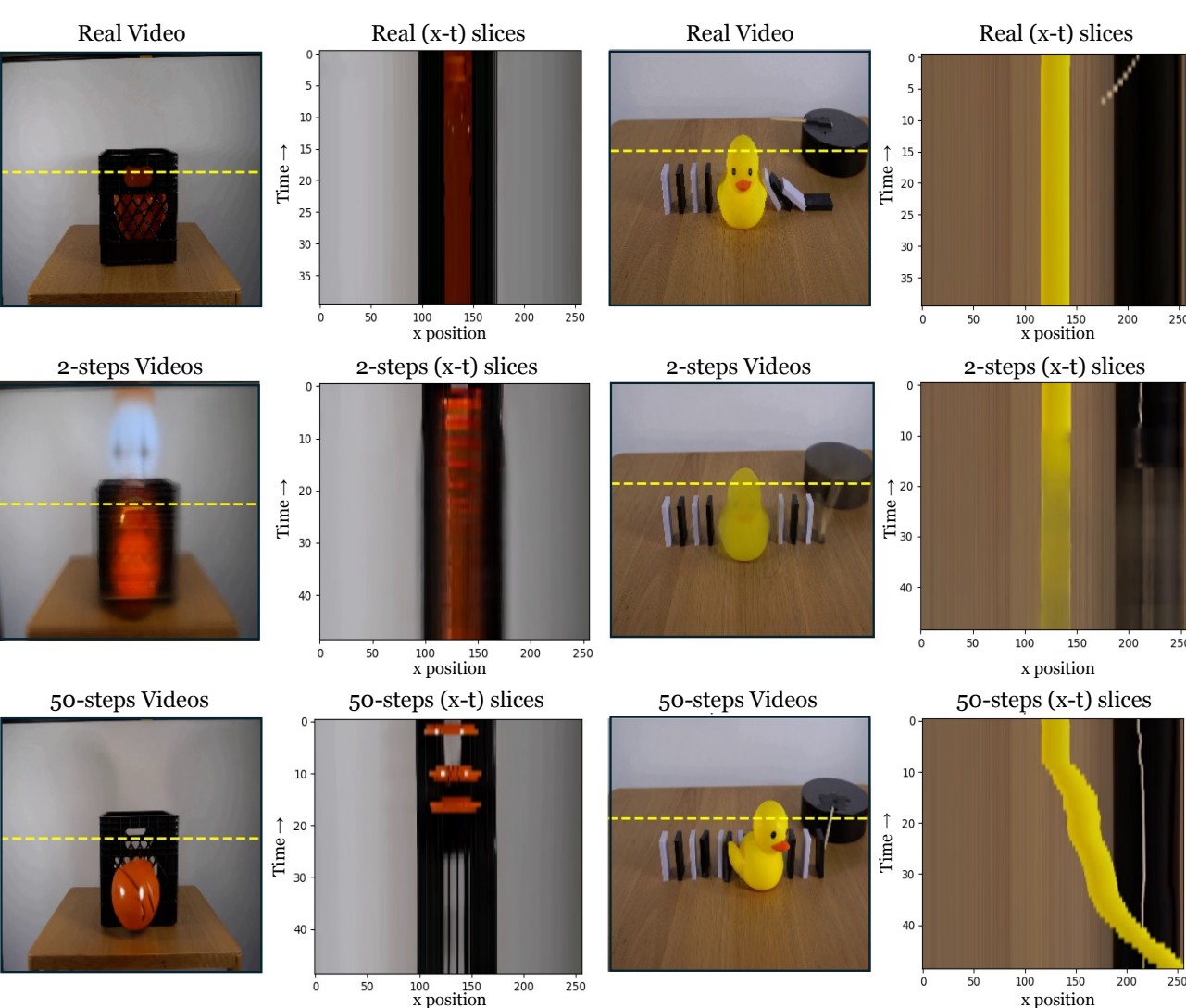

*Figure 16.* **Visualization of motion trajectories using spatio-temporal ($x$-$t$) slices**

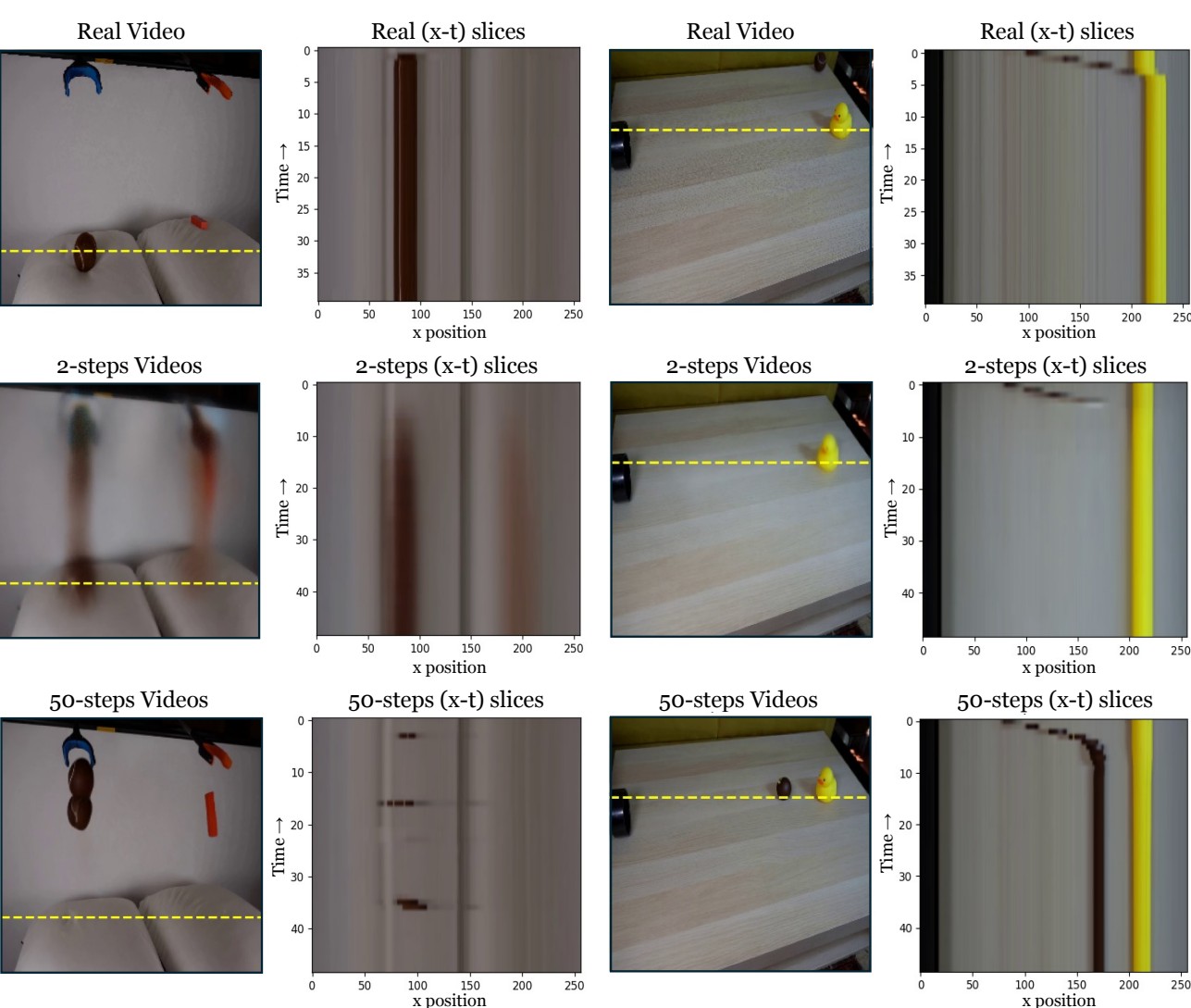

*Figure 17.* **Visualization of motion trajectories using spatio-temporal (*x*-*t*) slices**

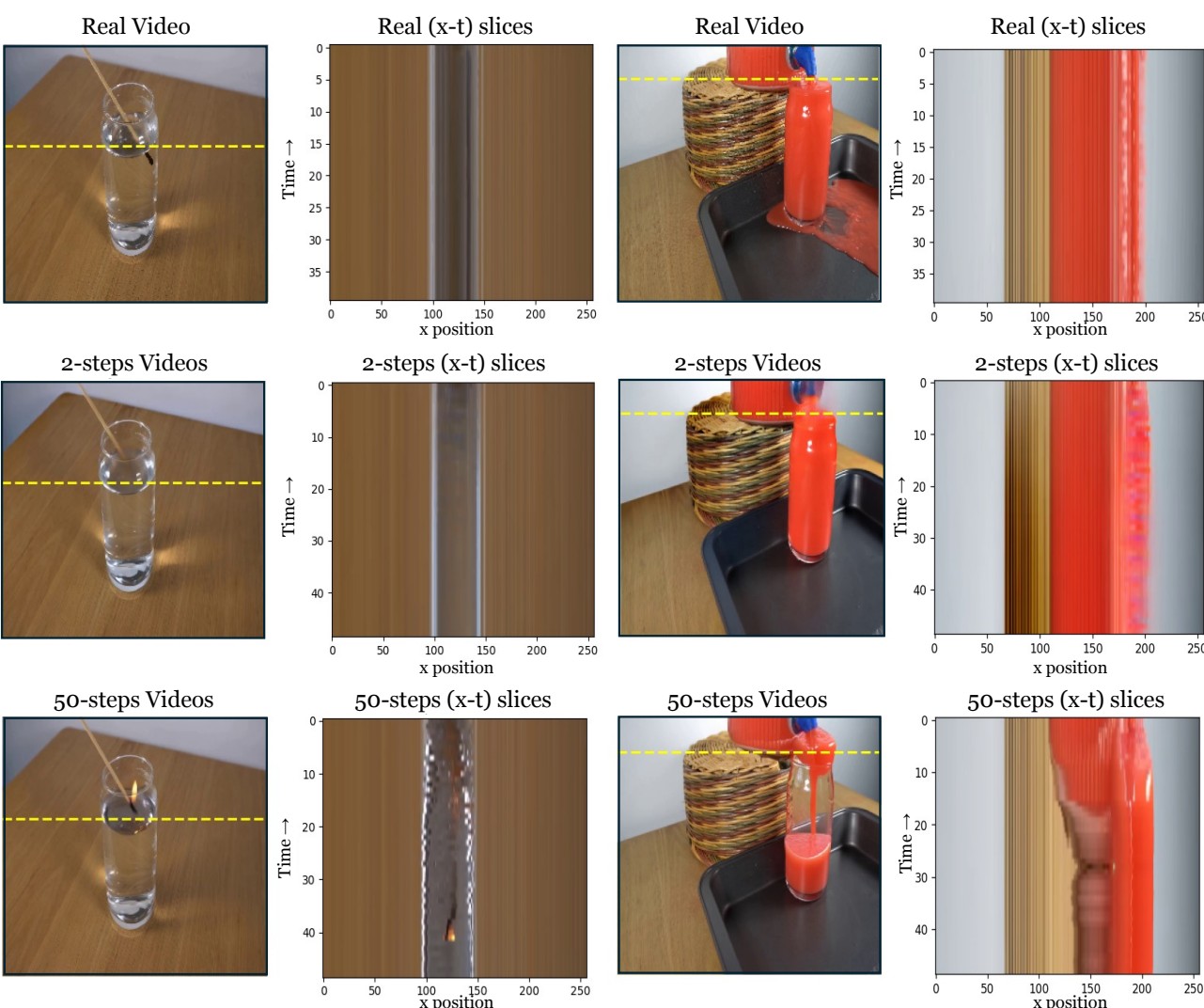

*Figure 18.* **Visualization of motion trajectories using spatio-temporal (*x-t*) slices**

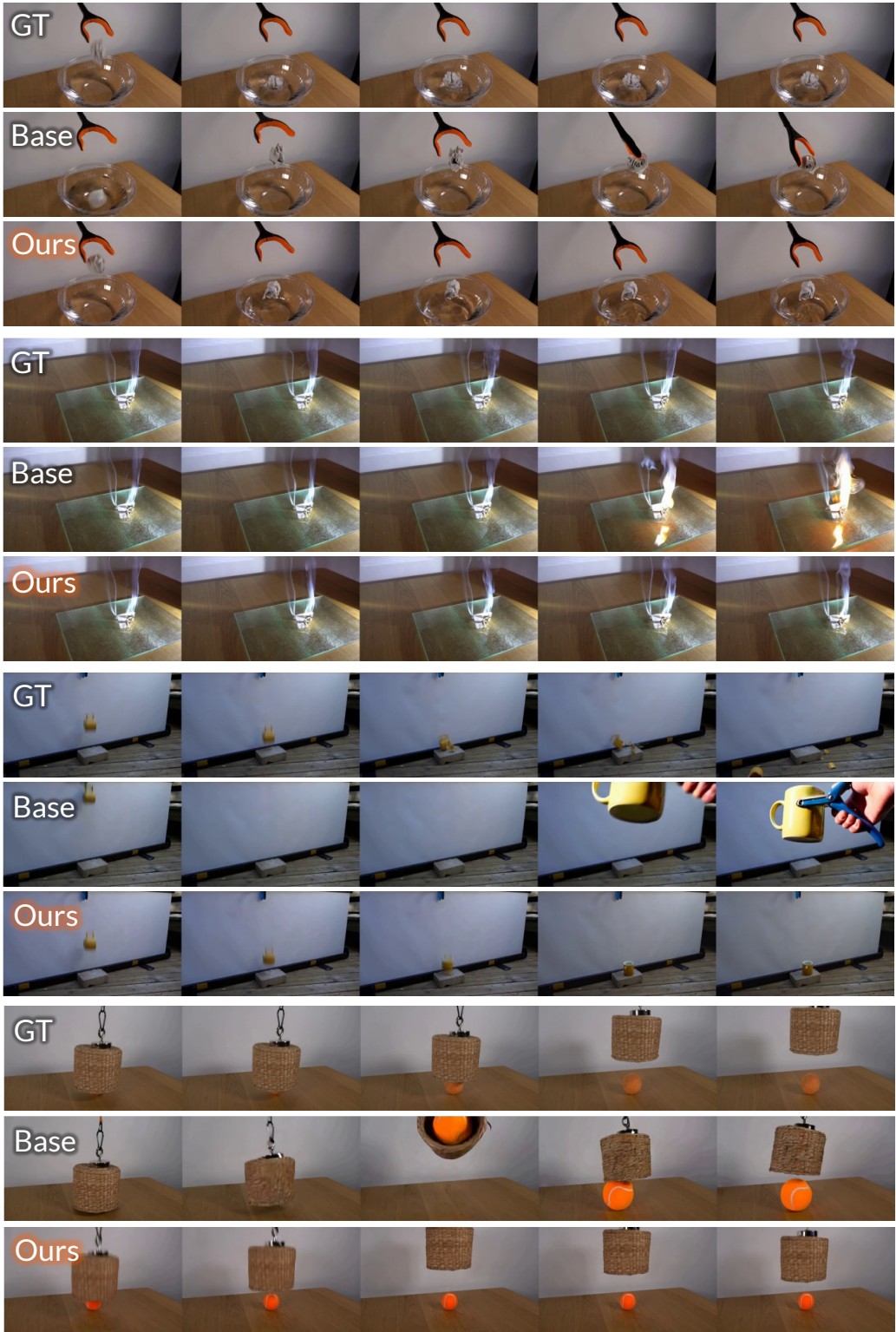

*Figure 19.* **Additional Qualitative Results for Physics-IQ Benchmark**

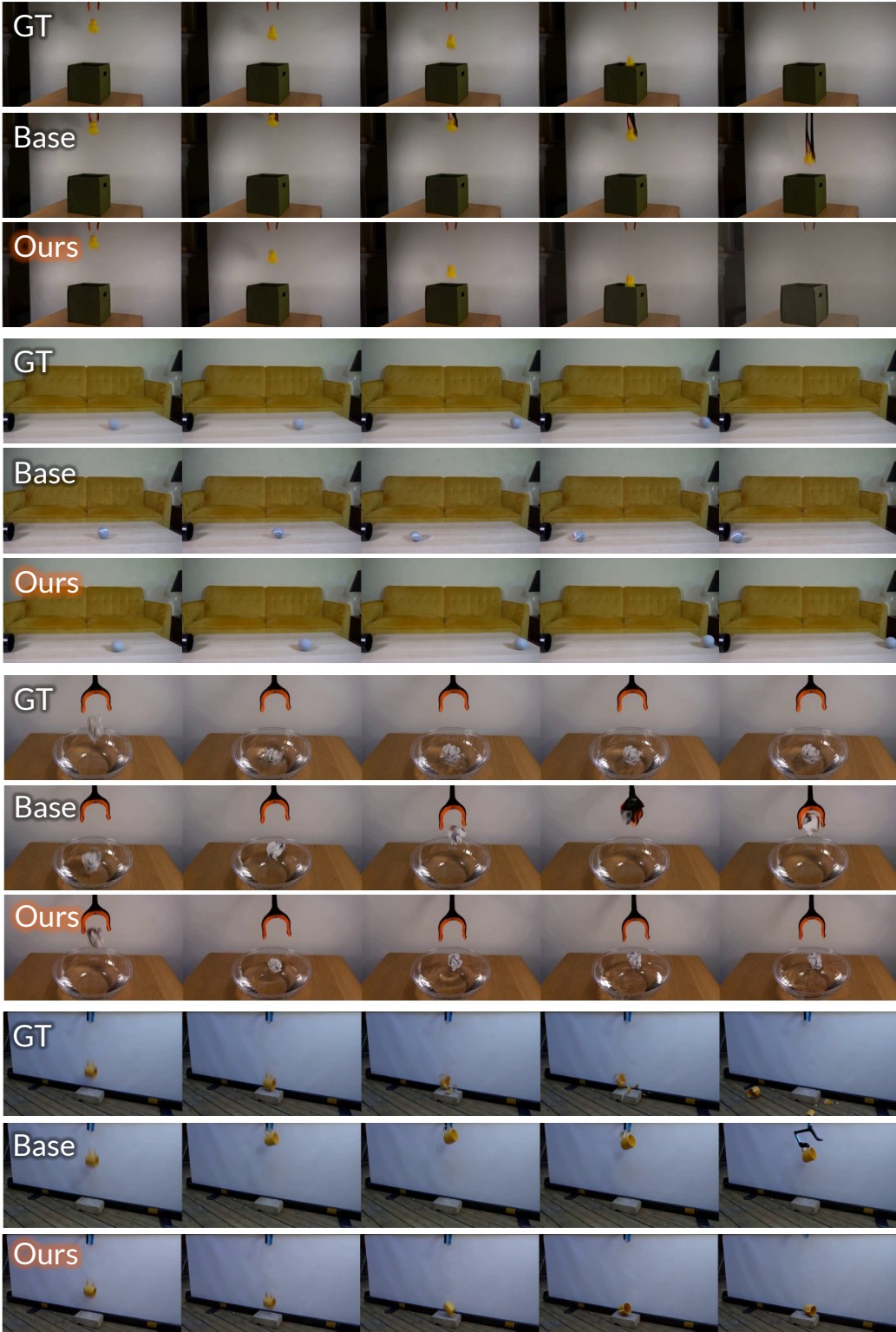

*Figure 20.* **Additional Qualitative Results for Physics-IQ Benchmark**

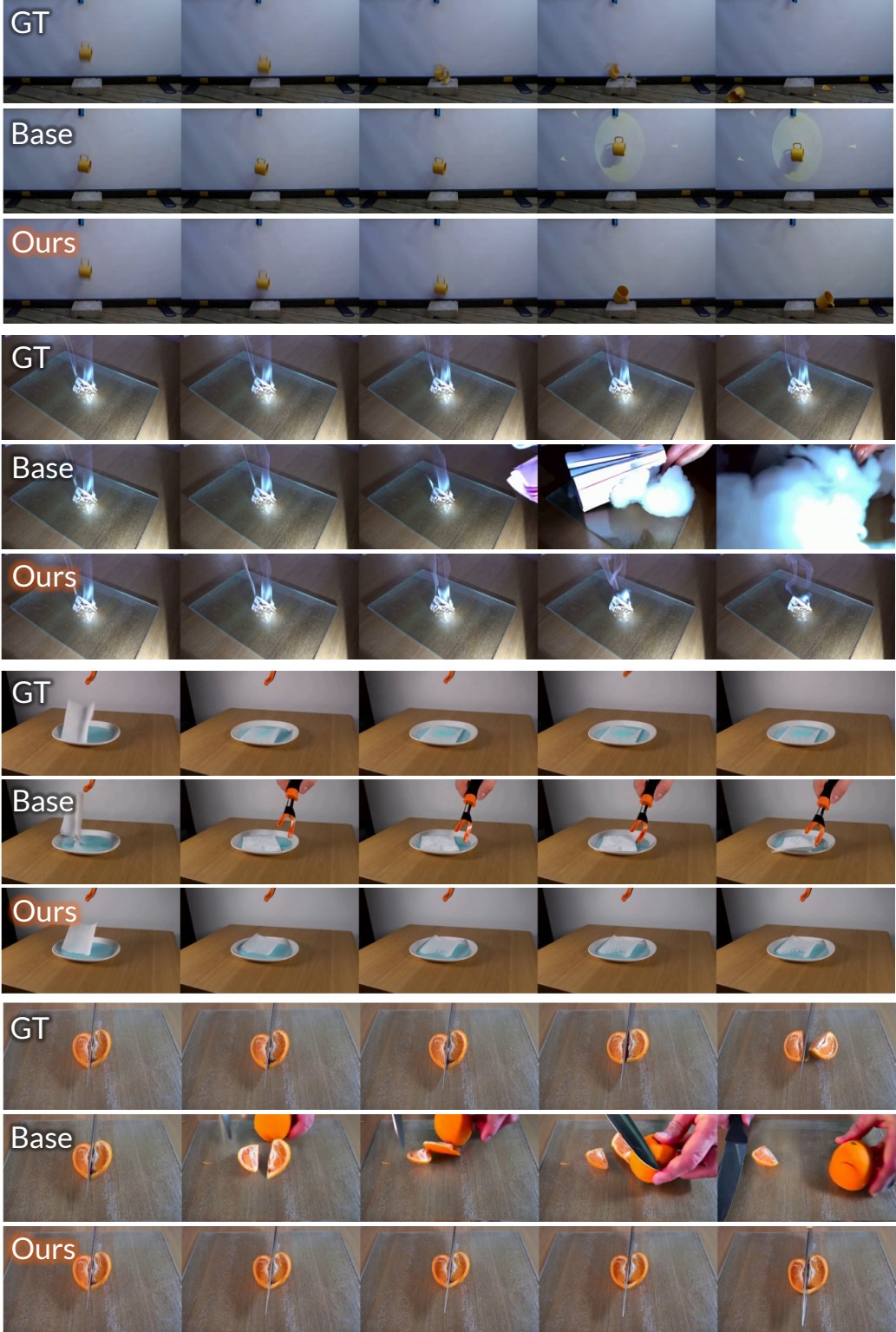

*Figure 21.* **Additional Qualitative Results for Physics-IQ Benchmark**

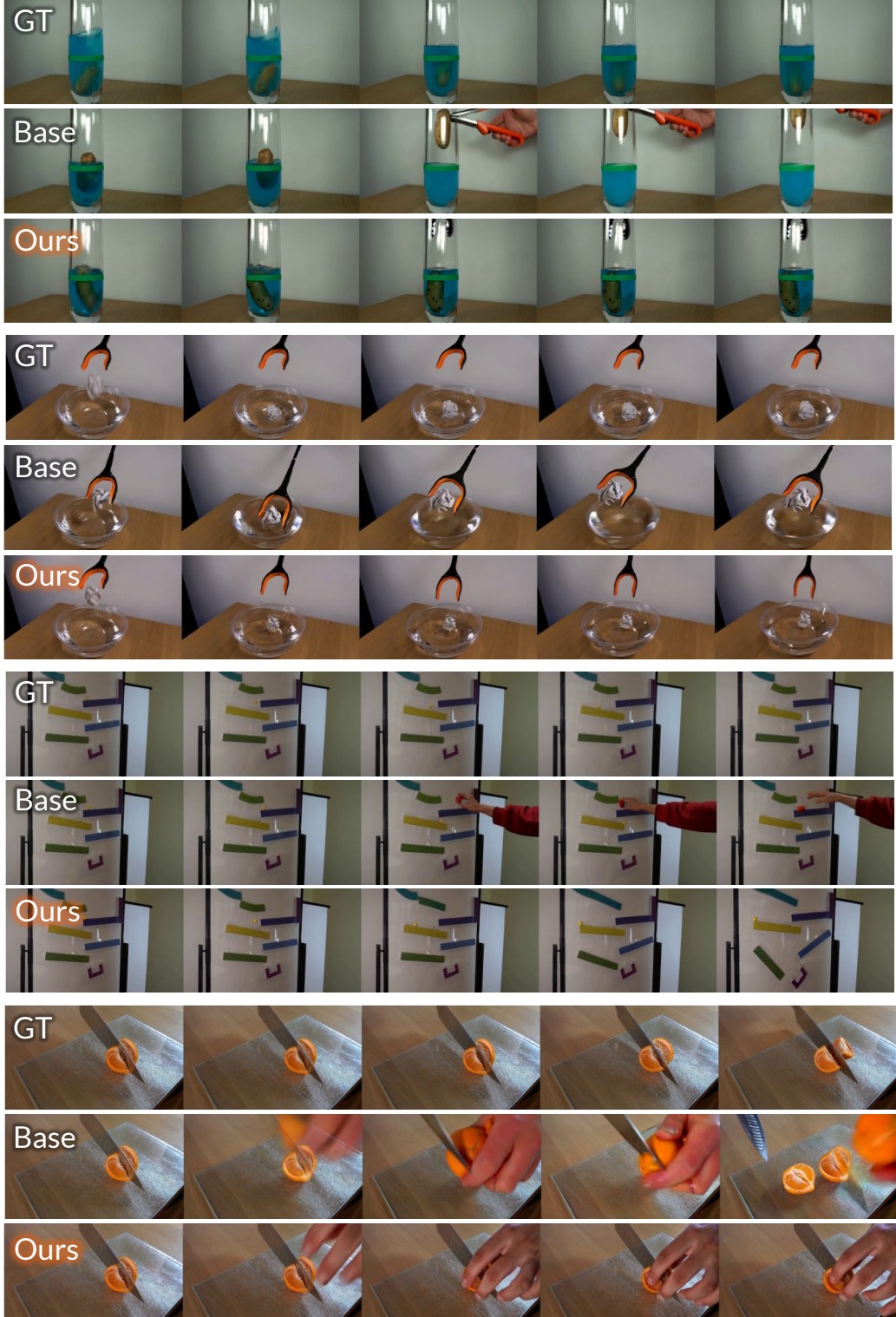

*Figure 22.* **Additional Qualitative Results for Physics-IQ Benchmark**

**Input Image**

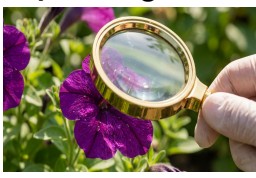

**Input Text**

A magnifying glass is gradually moving closer to the petals of a flower, revealing the intricate details and textures of the flower as it approaches.

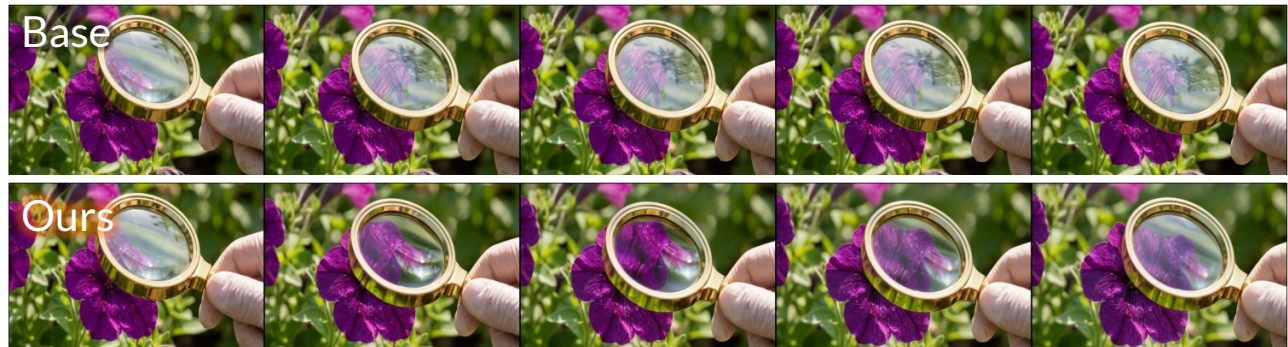

**Input Image**

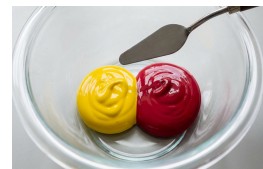

**Input Text**

Equal amounts of yellow and red paint are rapidly combined, with the mixture being vigorously stirred until fully blended.

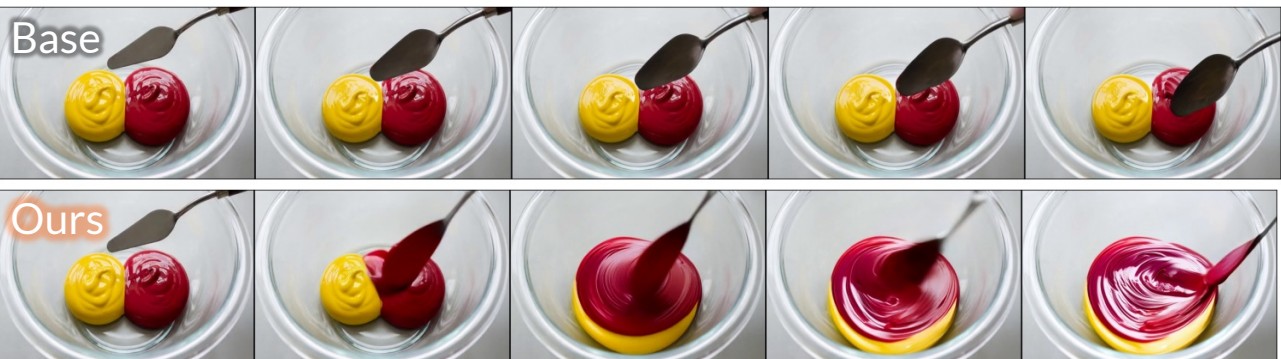

*Figure 23.* **Additional Qualitative Results for PhyGenBench**

**Input Image**

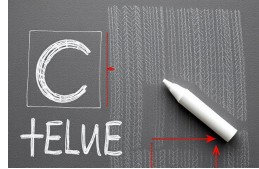

**Input Text**

A piece of white chalk is used to write on the rough, dark surface of a blackboard,showcasing the interaction between the chalk and the blackboard surface.

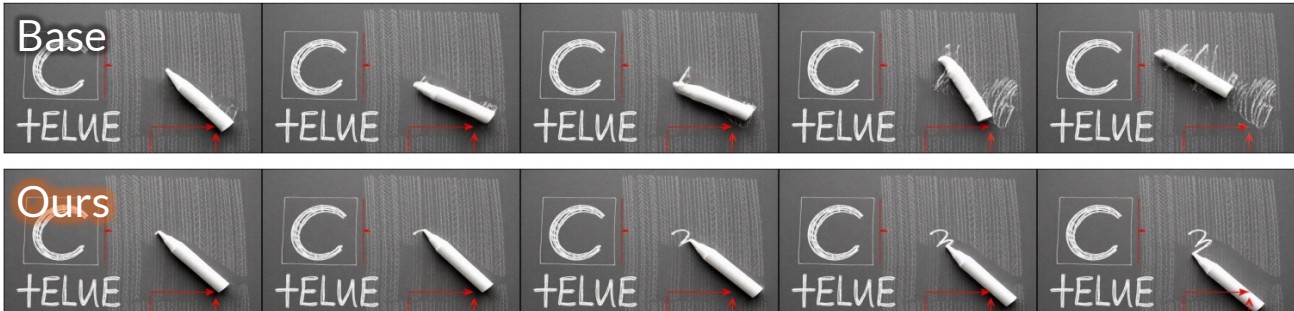

**Input Image**

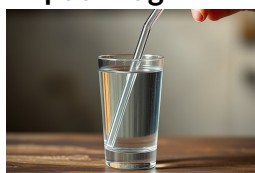

**Input Text**

A clear plastic straw is slowly inserted into a glass of crystal-clear water, revealing the fascinating visual changes and reflections that occur as the straw interacts with the liquid.

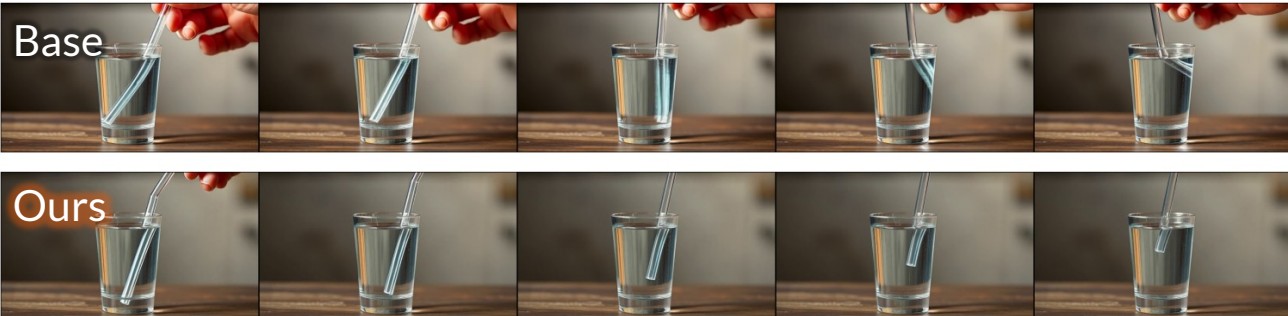

*Figure 24.* **Additional Qualitative Results for PhyGenBench**

**Input Image**

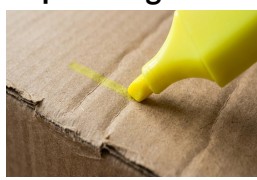

**Input Text**

A yellow highlighter is used to mark on the rough, brown surface of a cardboard, showcasing the interaction between the highlighter and the cardboard surface.

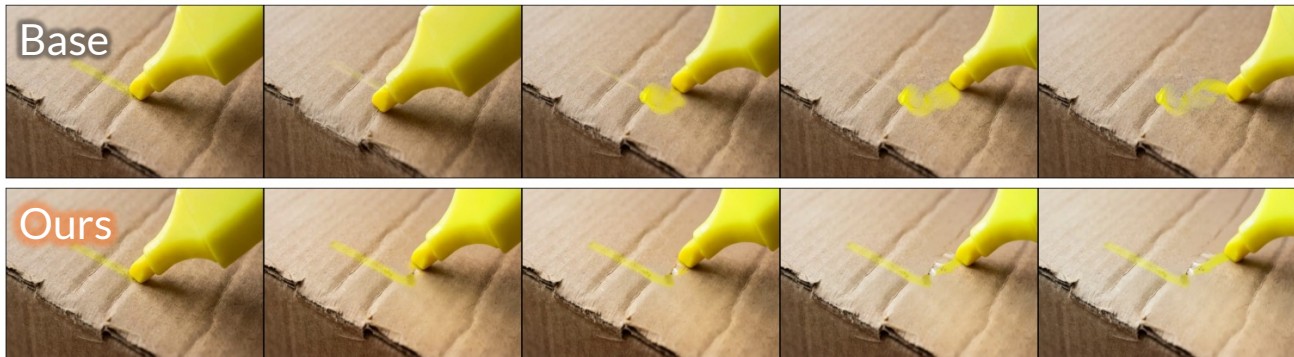

**Input Image**

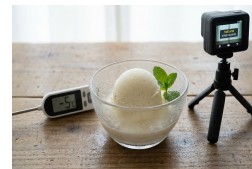

**Input Text**

A timelapse captures the gradual transformation of ice cream as the temperature rises significantly.

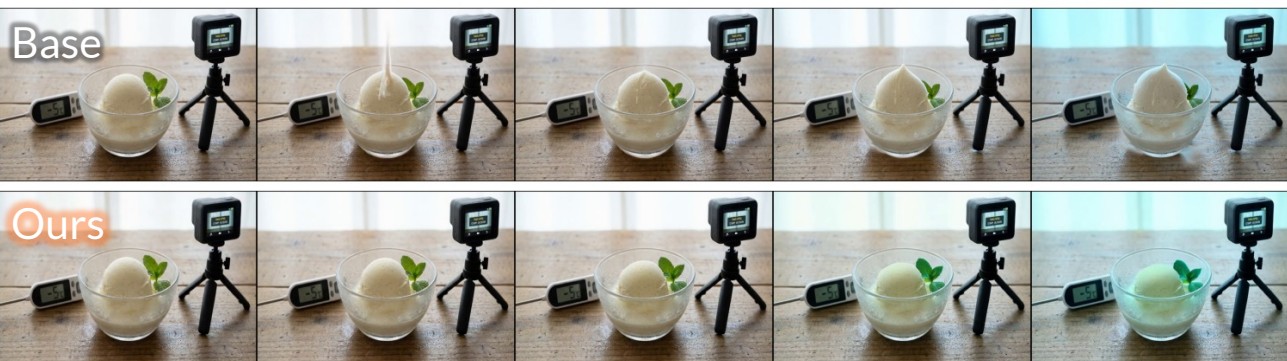

*Figure 25.* **Additional Qualitative Results for PhyGenBench**

**Input Image**

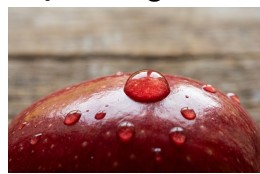

**Input Text**

A glistening dewdrop is sliding gracefully across the smooth surface of a waxed apple, accentuating its shape as it moves

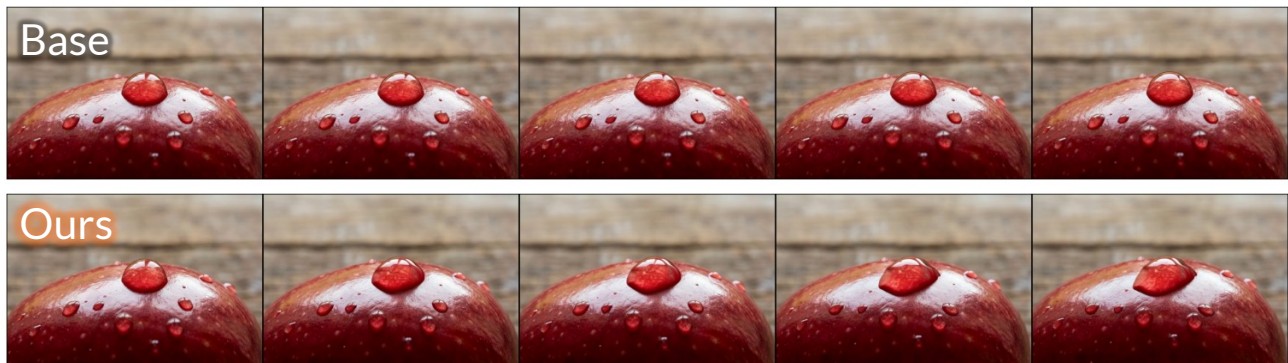

**Input Image**

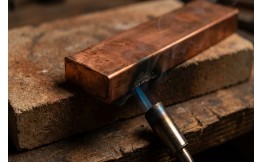

**Input Text**

A piece of copper is ignited, emitting a vivid and unique flame as it burns steadily.

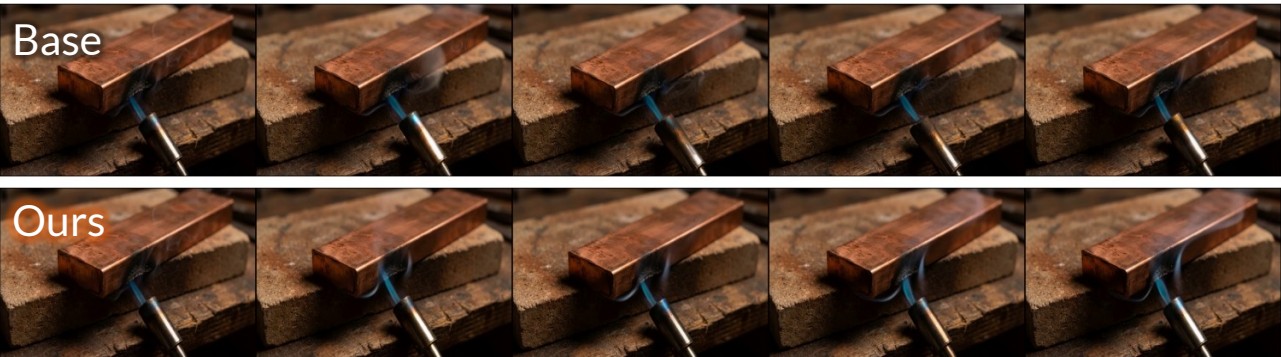

*Figure 26.* **Additional Qualitative Results for PhyGenBench**

