# OpenReview forum: "Physics in 2-Steps: Locking Motion Priors Before Visual Refinement Erases Them"
_ICML.cc/2026/Conference — ICML 2026 regular_

### Official Review · Reviewer_LAv9 · 2026-03-07

**Soundness:** 3
**Presentation:** 4
**Significance:** 4
**Originality:** 4
**Overall Recommendation:** 4
**Confidence:** 4

**Summary:**

This paper first presents an experimental finding that the physics fidelity of generated videos degrades as more NFEs are used. Based on this finding, they propose an inference-time method that maintains the physics accuracy of few-step videos while also achieving the visual realism of many-step models.

**Compliance With Llm Reviewing Policy:**

Affirmed.

**Key Questions For Authors:**

- Due to the aforementioned reasons, I think it would be helpful to provide a proper explanation for why physics realism can be better in videos with low NFEs. I think the high-level explanation in Appendix G.2.1 is heading in the right direction, but would it be possible to give a more detailed analysis with more rigor? It would be nice to include it in the main paper so that readers can be convinced that this phenomenon can actually exist. If space is a concern, maybe the current Section 4.4, Theoretical Justification, can be moved to the appendix.

- Can we isolate physics accuracy from prompt adherence?

**Limitations:**

yes

**Strengths And Weaknesses:**

Strength:

* The experimental finding that the physics fidelity of generated videos degrades as more NFEs are used is very interesting and puzzling.
* The proposed method is simple and does not require retraining.

Weakness

* As the authors acknowledge in Appendix G.2.1, they treat this phenomenon as a given and do not explain why it happens. This leads to the following question.

* The physics fidelity measures considered in this paper, such as Physics-IQ or Phase Coherence, mix prompt adherence with actual physics fidelity. Since both the motivation and the validation of the proposed method rely solely on numerical experiments, the validity of the work can be questioned.

* Specifically, I checked the supplementary videos on the provided project page and found that, for some videos, the baseline scores are low not because the generated videos have broken physics, but rather because they do not adhere to the prompt. For example,

* In "A grey tennis ball rolls out of the black tube", the baseline video might score low because the black tube vacuums the ball at the beginning, before it is launched, which it does twice. Other than that, the rolling motion of the ball itself looks fine.

* In "A piece of folded paper is placed on a glass cutting board", the baseline model generates a big rolled paper, which is not what the prompt asked for, and this may account for a large deviation from the ground-truth continuation.

Generating physically plausible videos and generating videos that adhere to the prompt are two different things. I do not think this confounding factor is large enough to overturn the result reported in this paper, as many other videos do show physics accuracy differences, but at the same time, it is hard to get a sense of how much gain can be attributed to improved physics realism.

---

> ### Author Rebuttal · Authors · 2026-03-31
>
> We sincerely thank the reviewer for the careful observation of our results and the insightful question.
>
> ## Q1. Isolating physics accuracy from prompt adherence
> We appreciate the reviewer reviewing our supplementary videos; we agree that in specific edge cases like the "black tube" or "folded paper", base model prompt failures can conflate visual errors with physical ones. However, our isolated human evaluations confirm that the gains are not reducible to prompt adherence.
> Specifically, our Human Study (Table 4) was conducted on the same Physics-IQ benchmark videos the reviewer examined, with five annotators scoring each video pair (Ours vs. Baseline) across three independently judged dimensions: (1) Physics Plausibility, (2) Visual Quality, and (3) Prompt Alignment. Because these judgments were made on the identical videos whose Physics-IQ scores the reviewer questions, the study directly isolates the confound within the same evaluation set.
>
> | | Physics Win | Prompt Win | Gap |
> |---|---|---|---|
> | vs. CogVideoX | 72.9% (Acc 58.0%) | 62.6% (Acc 54.1%) | +10.3%p |
> | vs. WAN 2.1 | 81.9% (Acc 67.1%) | 78.4% (Acc 63.1%) | +3.5%p |
>
> If the Physics-IQ gains were primarily driven by better prompt adherence, we would expect the Physics and Prompt win rates to be comparable. Instead, the Physics win rate consistently exceeds the Prompt win rate across both baselines, indicating a distinct gain in physical plausibility beyond what prompt adherence alone can explain.
>
> ## Q2. Why low NFE Exhibit Better Physics Realism?
> We sincerely thank the reviewer for this constructive suggestion. We present a rigorous analysis in four steps and will incorporate it into the main paper (moving Section 4.4 to the appendix).
>
> ### 1. Physical Motion Is Captured at Low NFE
> It is well established that diffusion models generate images in a coarse-to-fine manner: global structure (low frequencies) is resolved in early denoising steps, while fine-grained detail (high frequencies) emerges later. As Dieleman (2024) formalizes, this amounts to spectral autoregression in the reverse process. Since physical motion — object trajectories, velocity profiles — is predominantly a low-frequency phenomenon, the 2-step output already encodes the model's best estimate of global motion dynamics. The physical motion prior is not missing; it is the dominant low-frequency structure established in the earliest steps.
>
> ### 2. Phase Is More Vulnerable Than Magnitude Under L2 Training
> All major video diffusion models (noise prediction, v-prediction, flow matching) share an L2 training loss $\|\hat{\mathbf{y}} - \mathbf{y}\|_2^2$. By Parseval's theorem, this decomposes into per-frequency contributions. For a single frequency with true value $Ae^{j\phi}$ and prediction $\hat{A}e^{j\hat{\phi}}$:
> $$\frac{\partial \mathcal{L}}{\partial \hat{\phi}} = -2A\hat{A}\sin(\phi - \hat{\phi}), \quad \frac{\partial \mathcal{L}}{\partial \hat{A}} = 2(\hat{A} - A\cos(\phi - \hat{\phi}))$$
> The phase gradient is modulated by $A\hat{A}$, which vanishes at high frequencies where $A \propto f^{-1}$. The model is thus trained with weaker phase-correction than magnitude-correction capabilities, especially at higher frequencies.
> This describes the loss landscape, not inference dynamics directly, the network outputs pixel/latent values, not $A$ and $\phi$ independently. Nevertheless, the asymmetry shapes what the model learns, and empirical measurements confirm: phase coherence drops ~18% from step 2 to 50, while magnitude correlation drops only ~2–3%.
>
> ### 3. Phase Degradation → Physical Inconsistency
> As we derive in Appendix C (Eq. 5), the magnitude of the latent delta $|\mathcal{F}(\boldsymbol{\Delta})| \propto |\phi_f - \phi_{f-1}|$ , i.e., inter-frame motion is directly encoded as inter-frame phase differences. The phase vulnerability established above therefore directly manifests as corrupted motion dynamics. Our causal ablation (Section 3) confirms this link: 50% phase corruption causes $8.5\times$ greater optical flow distortion than equivalent magnitude corruption (EPE: 9.74 vs. 1.14).
> Taken together, low-NFE inference preserves physics not despite fewer steps, but because it limits exposure to the denoising process in which phase is structurally more vulnerable than magnitude.

---

> > ### Author Rebuttal · Reviewer_LAv9 · 2026-04-04
> >
> > The user study partially addresses my concern, but it's not very convincing since there are only five annotators. I maintain my score.

---

> > > ### Author Response · Authors · 2026-04-05
> > >
> > > Thank you very much for your feedback and follow-up comments.
> > > We deeply appreciate the time you took to carefully review our paper and provide constructive questions. Your insights have helped us significantly clarify the theoretical mechanism behind physics preservation at low NFEs and the validity of our evaluation metrics of our work.
> > >
> > > In response to your latest comments, we have provided further clarifications and conducted additional experiments.
> > >
> > > For context, our user study was conducted using the exact same settings as the prior work ([1] WMReward). We ensured **there was no cherry-picking by evaluating all videos** from the Physics-IQ Benchmark. Furthermore, during the study, the video pairs were **randomly shuffled** for comparison, and the results were only revealed at the end. While we were only able to recruit 4 additional participants due to the short rebuttal timeframe, we commit to recruiting a larger pool of participants for the final version. Additionally, we have shared the resulting videos in the supplementary material and will ensure they are included with the final version.
> > >
> > > We would like to highlight that the win rate for Physics Plausibility consistently outperforms Prompt Alignment by a significant margin. It proves that our method provides substantial gains in physical realism that cannot be attributed solely to better prompt adherence.
> > >
> > > **[Ours vs. CogVideoX]**
> > >
> > > | Criteria | Ours (Win) | Ours (Acc) | CogVideoX (Win) | CogVideoX (Acc) |
> > > | :--- | :---: | :---: | :---: | :---: |
> > > | Physics Plausibility | **77.2** | **58.0** | 22.7 | 42.0 |
> > > | Visual Quality | **78.7** | **62.3** | 21.3 | 37.7 |
> > > | Prompt Alignment | **59.7** | **53.8** | 40.3 | 46.2 |
> > >
> > >
> > > **[Ours vs. WAN 2.1]**
> > >
> > > | Criteria | Ours (Win) | Ours (Acc) | WAN 2.1 (Win) | WAN 2.1 (Acc) |
> > > | :--- | :---: | :---: | :---: | :---: |
> > > | Physics Plausibility | **84.9** | **73.1** | 15.1 | 26.9 |
> > > | Visual Quality | **88.0** | **75.6** | 12.0 | 24.4 |
> > > | Prompt Alignment | **78.9** | **66.4** | 21.1 | 33.6 |
> > >
> > > Thank you once again for your time, interest, and continued support!
> > >
> > > [1] Yuan, Jianhao, et al. "Inference-time Physics Alignment of Video Generative Models with Latent World Models." arXiv preprint arXiv:2601.10553 (2026).

---

### Official Review · Reviewer_iC5c · 2026-03-12

**Soundness:** 3
**Presentation:** 4
**Significance:** 3
**Originality:** 3
**Overall Recommendation:** 4
**Confidence:** 4

**Summary:**

The paper presents a compelling and counter-intuitive discovery: videos generated by diffusion models with minimal inference steps exhibit superior physical consistency compared to standard 50-step generations. Furthermore, the experimental validation is extensive, supported by a comprehensive quantitative analysis and well-considered evaluation metrics. The proposed method also demonstrates strong generalization capabilities.

**Compliance With Llm Reviewing Policy:**

Affirmed.

**Final Justification:**

The authors have addressed my concerns.

**Key Questions For Authors:**

1-Limited Discussion on Non-Rigid Dynamics: There is insufficient discussion regarding non-rigid scenarios (e.g., cloth folding, soft-body floating). It remains unclear whether PhaseLock might fail to represent complex, non-linear physical laws in these contexts. More extensive experiments on non-rigid scenes are encouraged.
2-Handling of High-Frequency and Instantaneous Dynamics: For physical dynamics involving high-frequency collisions or fracturing—which involve instantaneous changes and sharp boundaries—simply locking low-frequency phases may not suffice to restore complex interactions. The lack of analysis in these scenarios raises concerns regarding the method's robustness.
3-Generalizability of NFE Settings: Regarding the Number of Function Evaluations (NFE) settings under the Motion Prior, the experiments primarily default to 50 steps. It is unclear if these NFE settings remain applicable for models designed with inherently low recommended steps (e.g., one-step or few-step video models).

**Limitations:**

Yes

**Strengths And Weaknesses:**

Strengths：
1-Novel and Counter-intuitive Insight: The paper presents a compelling and counter-intuitive discovery: videos generated by diffusion models with minimal inference steps exhibit superior physical consistency compared to standard 50-step generations. This suggests that while diffusion models possess internal physical priors, these are inadvertently disrupted during the visual refinement process.
2-Rigorous Theoretical Grounding: The authors provide a solid quantitative analysis identifying "Phase Erosion" as the root cause of declining physical consistency. This provides a firm theoretical foundation for the proposed solution.
3-Practicality and Generalization: PhaseLock is a training-free method, which significantly enhances its ease of implementation and potential for broad generalization across different frameworks.

Weaknesses：
1-Limited Discussion on Non-Rigid Dynamics: There is insufficient discussion regarding non-rigid scenarios (e.g., cloth folding, soft-body floating). It remains unclear whether PhaseLock might fail to represent complex, non-linear physical laws in these contexts. More extensive experiments on non-rigid scenes are encouraged.
2-Handling of High-Frequency and Instantaneous Dynamics: For physical dynamics involving high-frequency collisions or fracturing—which involve instantaneous changes and sharp boundaries—simply locking low-frequency phases may not suffice to restore complex interactions. The lack of analysis in these scenarios raises concerns regarding the method's robustness.
3-Generalizability of NFE Settings: Regarding the Number of Function Evaluations (NFE) settings under the Motion Prior, the experiments primarily default to 50 steps. It is unclear if these NFE settings remain applicable for models designed with inherently low recommended steps (e.g., one-step or few-step video models).
4-Model Diversity: To further validate the method's value, it should be tested across a wider variety of video generation architectures beyond the current selection.
5-Comparative Analysis: The paper would benefit from a broader comparison with physically-guided models. This would better highlight the advantages of a "guidance-free" approach compared to methods requiring external physical constraints.

---

> ### Author Rebuttal · Authors · 2026-03-31
>
> We thank the reviewer for recognizing our novelty, theoretical grounding, and generalization. We address each weakness with new quantitative evidence. All referenced figures and tables are at: https://anonymous.4open.science/r/12936.
>
> ## W1: Non-rigid dynamics
>
> PhaseLock improves non-rigid scenarios by **+41.8%**, exceeding rigid gains (+23.4%), suggesting the method is especially suited to complex deformable dynamics. Of the 66 Physics-IQ scenarios, 27 (41%) involve non-rigid dynamics: fluid (15, e.g., pouring, capillary flow, siphon), deformable solids (9, e.g., cloth draping, soft-body cutting, granular deformation), and thermodynamics (3, e.g., combustion, balloon rupture). The per-scenario breakdown is in `Table 1`.
>
> Representative gains include: `napkin-soak` (+54.7 pts), `siphon` (+55.4 pts), `cut-orange` (+51.9 pts), and `silk-cover` (+35.8 pts on CogVideoX).
>
> In PhyGenBench, Thermal and Material Properties categories (melting, dissolution, combustion) show clear improvements in both (`Table 2`), with Material Properties achieving +36.7%.
>
> We demonstrate cloth folding and soft-body floating in `Figure 1`.
>
> ## W2: High-frequency and instantaneous dynamics
>
> Latent Delta Guidance operates in the full spatial domain; by Parseval's theorem, it aggregates over all frequencies simultaneously, including high-frequency components encoding collisions and sharp boundaries. The paper's "low-frequency" framing describes where we *diagnose* phase erosion (Sec. 3.2), not the operating range of guidance.
>
> The empirical consequence is visible in the exact scenario types the reviewer describes: fracture/shattering (Physics-IQ Solid Mechanics subset), sharp-boundary cutting (`cut-orange`: +51.9 pts), rapid fluid transitions (`siphon`: +55.4 pts), and combustion (`match-blows-balloon`). On PhyGenBench, Material Properties shows our largest relative gain (+36.7%).
>
> Regarding instantaneous dynamics: guidance targets the velocity field (inter-frame deltas), not absolute phase values. Even when a collision creates a spatial discontinuity, the 2-step prior captures the direction and onset of that event. The adaptive linear decay reinforces this: guidance is strongest in early steps when event trajectories are established, then relaxes to allow high-frequency detail rendering. PhaseLock thus functions precisely where discontinuous physics are hardest to model.
>
> ## W3: Generalizability of NFE settings
>
> A new experiment on WAN 2.1 Distilled (LightX2V, 4-step) with K_fast=2, K_full=4 confirms generalization (`Table 3`): Physics-IQ improves from 27.7 to 29.4.
>
> The smaller gain (+1.7 vs +7.8) is mechanistically expected, not a failure. Fewer denoising steps create fewer opportunities for phase erosion, so the distilled baseline already achieves 27.7, substantially exceeding the full 50-step baseline (20.9). This directly confirms our core finding: fewer steps better preserve physics. The smaller improvement is confirmatory evidence, not a limitation.
>
> ## W4: Model diversity
>
> We have tested on three additional architectures since submission (`Table 4`):
>
> - **SVD** (UNet-based): +1.2, confirms our approach is not limited to Transformer-based generators
> - **SkyReels-V2** (different scheduler and resolution pipeline): +8.9, largest absolute gain
> - **WAN 2.1 Distilled** (4-step DiT): +1.7, compatible with accelerated pipelines
>
> These span UNet, Expert Transformer, Lightweight DiT, and standard DiT, covering both full-step and distilled inference. The SVD gain (+1.2) is smaller, consistent with UNet-based models exhibiting structurally less inter-frame phase coupling in latent space. This coverage demonstrates architectural independence of the core mechanism.
>
> ## W5: Comparative analysis
>
> We provide a systematic comparison in `Table 5`. The reviewer is right that this comparison is needed; without it, the "guidance-free" advantage is asserted rather than demonstrated.
>
> **(1) Direct quantitative comparison.** PhaseLock scores 36.0/36.3 = **99.2%** of WMReward's performance. WMReward(∇+BoN) costs ×5.02×N; at N=16 (Fig. 7) this yields ≈**80× more computation** on the same H100, plus external Video-JEPA models and gradient backpropagation. PhaseLock is strictly training-free and gradient-free.
>
> **(2) Why simulation methods are not directly comparable.** PhysGen and PhysCtrl require manual per-scenario specification of force vectors, torque, mass, and friction, supporting only rigid-body dynamics. These cannot be applied to fluid, thermodynamics, or deformable-solid scenarios without per-scenario expert curation, inapplicable at Physics-IQ's 66-scenario scale. WMReward is thus the only methodologically valid automated comparison.
>
> **(3) Complementary positioning.** PhaseLock preserves intrinsic physical knowledge rather than injecting external physics, making it combinable with training-based methods, e.g., applying PhaseLock on a VideoREPA-finetuned backbone could yield cumulative gains.

---

> > ### Author Rebuttal · Reviewer_iC5c · 2026-04-03
> >
> > The authors have addressed my concerns.

---

> > > ### Author Response · Authors · 2026-04-07
> > >
> > > We are glad to have successfully addressed your concerns. We truly appreciate the constructive feedback throughout the review process, as your insightful feedback has made our work more solid.
> > >
> > > All the new analyses and experiments will be incorporated into the final version of the manuscript.

---

### Official Review · Reviewer_W8Tm · 2026-03-12

**Soundness:** 2
**Presentation:** 2
**Significance:** 1
**Originality:** 2
**Overall Recommendation:** 4
**Confidence:** 4

**Summary:**

The authors present a training-free framework called PhaseLock to improve the physical consistency of video diffusion models. The core premise is based on the observation that a 2-step generation often exhibits better physical consistency than a 50-step output due to phase erosion during the denoising process. PhaseLock extracts a motion prior from a few-step inference and enforces it on the high-fidelity generation via Latent Delta Guidance.

**Compliance With Llm Reviewing Policy:**

Affirmed.

**Final Justification:**

My concerns have been addressed.

**Key Questions For Authors:**

How can the framework accurately detect and recover from an inherently flawed 2-step motion prior before propagating it through the remaining denoising steps?

Given the catastrophic failure of the direct latent guidance baseline, how sensitive is the Latent Delta Guidance to subtle structural misalignments between the 2-step prior and the intended high-fidelity output?

**Limitations:**

Yes

**Strengths And Weaknesses:**

**Strengths**

The paper provides an interesting spectral analysis of the generation process, highlighting that phase degrades significantly while magnitude remains stable.

The proposed PhaseLock method is training-free and introduces minimal computational overhead (1.06x time), circumventing the need for expensive external guidance methods.

**Weaknesses**

The foundational reliance on the 2-step generation is a critical vulnerability. The authors acknowledge that if the 2-step generation produces incorrect physics—whether due to ambiguous inputs or intrinsic limitations—the guidance strictly locks the high-fidelity generation to this erroneous trajectory. This severely impacts the robustness and reliability of the method in open-ended prompts. Furthermore, strong guidance can over-constrain the generation, preventing the model from properly refining high-frequency visual details.

The method is architecturally restricted. It relies entirely on the iterative denoising process and cannot be applied to autoregressive models, which limits its broader utility in the rapidly evolving landscape of video generation. Additionally, the method restricts creative or artistic motions that users might intentionally desire.

The paper is generally well-structured, and the spectral mechanism is clearly explained. However, the failure cases highlighted in the appendix are significant enough that they undermine the core thesis and should be more transparently addressed in the main text's evaluation.

The perspective of identifying phase erosion as the primary culprit for physical hallucinations and utilizing a spatial-domain proxy (latent delta) to lock phase dynamics is a creative approach to the problem.

---

> ### Author Rebuttal · Authors · 2026-03-31
>
> We thank the reviewer for recognizing the creativity of our approach. We address the weaknesses with new quantitative evidence. All referenced figures and tables are available at: https://anonymous.4open.science/r/12936.
>
> ## W1. Robustness of the 2-step prior
>
> To address concerns regarding robustness and over-constraining, we provide more context on how our guidance mechanism is designed to mitigate them.
> We conducted a per-scenario analysis across all 66 Physics-IQ scenarios (`Fig.1`), measuring spatiotemporal IoU. The 2-step prior is predominantly accurate: it improved motion dynamics in 74% (49/66) of scenarios for Wan 2.1 and 67% (44/66) for CogVideoX-5b, with maximum improvements (+55.4, +57.9) significantly outweighing maximum degradations (−20.3, −22.2).
> Regarding the concern that our method "strictly locks" the generation to erroneous trajectories: the guidance functions as a soft, decaying nudge, not a rigid constraint. At $\lambda_0=0.05$ with linear decay, at least 95% of the base model's denoising trajectory is preserved at every step. Our ablations (Sec. F) confirm this: across all tested configurations ($\lambda_0$, guide_end), the model consistently outperforms the unguided baseline (30.82).
> To prevent over-constraining, our method incorporates two safeguards: (1) guiding only inter-frame motion, not spatial features, and (2) halting guidance at step $K/2$ to leave late-stage detail refinement unimpeded. VBench and human evaluations validate that visual quality is preserved.
> We will add the per-scenario analysis to Sec. G.1 and soften "lock" to "guide" in the methodology.
>
> ## W2. Architectural Scope
>
> We acknowledge that PhaseLock targets diffusion-based pipelines. However, as shown in `Table 1`, we validated our method across six models spanning four distinct architectures (Expert Transformer, DiT, Lightweight DiT, UNet), all showing consistent improvements despite differences in attention mechanisms, temporal modeling, parameter counts, and inference paradigms (50-step vs. 4-step distilled). This suggests phase erosion is a general property of iterative refinement, not a quirk of one architecture. While phase erosion as we define it is specific to iterative refinement, the broader insight, that coarser outputs may preserve dynamics that finer outputs lose, could inform future work on other paradigms.
>
> ## W3. Restricted creative or artistic motions
> Our method does not impose external physical laws, it extracts the motion prior from the same model, same prompt, and same seed via 2-step inference. If a user prompts "a ball floating upward," the 2-step prior itself generates upward motion, and PhaseLock preserves that trajectory into the 50-step output. PhaseLock corrects cases where the model's own physical knowledge is erased during refinement, not cases where the user specifies non-physical motion. Furthermore, $\lambda_0$ provides continuous control ($\lambda_0=0$ recovers the exact baseline). In essence, we aim to preserve the inherent physics that are taken for granted even when left unsaid.
>
> ## W4. Failure cases
> We completely agree failure cases should be in the main text (move to Sec. 5 or 6) in the camera-ready. A detailed per-scenario breakdown with failure categorization will be included in Appendix G.1. That said, we want to emphasize that these failures do not undermine the overall contribution. As shown in our W1 analysis, PhaseLock improves 74% of scenarios (Wan 2.1) and 67% (CogVideoX-5b).
>
> ## Q1. Detecting flawed 2-step priors
> PhaseLock does not incorporate an explicit detection mechanism, as this would require an additional computationally heavy module. However, the framework is inherently robust to flawed priors: $\lambda_0=0.05$ preserves ~95% of the original trajectory, and our ablations (Sec. F) confirm that even at $3\times$ the default strength, performance stays at baseline level rather than collapsing. Guidance is also restricted to the first half of denoising ($k_{\text{end}}=K/2$), giving the model sufficient later steps to self-correct. Active detection (e.g., monitoring discrepancy between $M_{\text{prior}}$ and $M_{\text{current}}$) is a promising direction for future work.
>
> ## Q2. Sensitivity of Latent Delta Guidance to structural misalignments
> Latent Delta Guidance is structurally robust to this fidelity gap. Direct latent guidance ($\mathbf{z}^{\text{prior}} - \mathbf{z}^{(k)}$) fails (score 16.0) because the residual is dominated by appearance gap, not dynamics. Our delta operator $\mathcal{T}(\mathbf{z}) = \mathbf{z}\_{2:F} - \mathbf{z}\_{1:F-1}$ subtracts consecutive frames within the same path, projecting out time-independent content so only inter-frame motion remains. Three results confirm this: (1) under heavy blur $\sigma=16$, Step 2 still shows 3.6× higher phase correlation than Step 50 (Fig. 3a); (2) magnitude corruption has 8.5× less impact than phase corruption (Fig. 3b); (3) performance improves across all tested hyperparameter ranges (Sec.F)

---

> > ### Author Rebuttal · Reviewer_W8Tm · 2026-04-05
> >
> > N.A.

---

> > > ### Author Response · Authors · 2026-04-07
> > >
> > > We are glad to have successfully addressed your concerns. We truly appreciate the constructive feedback throughout the review process, as your insightful feedback has made our work more solid.
> > >
> > > All the new analyses and experiments will be incorporated into the final version of the manuscript.

---

### Official Review · Reviewer_eZVx · 2026-03-13

**Soundness:** 3
**Presentation:** 4
**Significance:** 3
**Originality:** 3
**Overall Recommendation:** 4
**Confidence:** 3

**Summary:**

This work proposes a training-free method for enhancing physical motion in I2V diffusion models. It first gives an analysis regarding the trade-off between visual refinement and physics consistency. Based on the analysis, it proposes PhaseLock, a method that utilizes the generation results of a 2-step method as a coarse motion prior. The coarse motion prior is used to guide full denoising steps through Latent Delta Guidance (i.e., a method for constraining latent deltas), allowing for significant increase in physics benchmarks with minimal additional overhead.

**Compliance With Llm Reviewing Policy:**

Affirmed.

**Final Justification:**

I thank the authors for their detailed rebuttal, which has resolved my concerns. Overall, I think the motivation and methodology are interesting and persuasive, and I will maintain my positive assessment of 4.

**Key Questions For Authors:**

1. Coarse motion priors are extracted using 2-step denoising (i.e., $K_{few}$=2). Ablation study over $K_{few}$ provided in Appendix F.1.3 shows that performance decreases sharply as the number of few-steps increases. Is this always the case for all scenarios, or could it vary model by model?
2. During motion prior extraction, the VAE takes in blurry frames as input. However, when the backbone diffusion model was initially trained, this VAE did not see such blurry frames, so these inputs are likely OOD from the perspective of the VAE. Could this cause the coarse latent sequence $z^{few}$ to be corrupted in any way?
3. Would it be possible to train the model to incorporate physics in a similar way (and not only correcting during inference time)?

**Limitations:**

Yes.

**Strengths And Weaknesses:**

### Strengths
1. The manuscript is well presented and the motivation for using a 2-step method as motion prior is persuasive. Analysis experiments like spatio-temporal slices (Figure 2a) and low-frequency spectral analysis (Figure 2b) shows concrete evidence for the methods taken in the paper.
2. Performance gains are quite impressive, especially considering that only a minimal amount of additional time/memory is needed for these gains.
3. The paper discusses alternative choices that could be made other than Latent Delta Guidance, and thoroughly explains the disadvantages of such naïve approaches in Appendix D.3. This supports the effectiveness and novelty of the proposed Latent Delta Guidance.

### Weaknesses
1. The entire method is built upon the assumption that "the 2-step generated motion prior contains accurate motion priors". In other words, if the 2-step generated motion prior is not accurate, the entire method could collapse and the video generation could actually be guided towards incorrect motion dynamics. The authors discuss this limitation in part in Appendix G.1, but since this is such a significant assumption I think it would be best if some more analysis could be provided regarding the frequency of the 2-step prior being inaccurate.
2. Theoretical justifications are built on the assumption that "consecutive frames typically share similar magnitude spectra". Thus, the method may work well for smooth motion but not for dynamic videos or videos with a lot of noise/motion in them.
3. The proposed method is only applicable for diffusion-based methods, but the authors discuss this in limitations and is just a minor weakness.

---

> ### Author Rebuttal · Authors · 2026-03-31
>
> We thank the reviewer for recognizing our novelty, theoretical grounding, and generalization. We address the weaknesses with new quantitative evidence. All referenced figures and tables are available at: https://anonymous.4open.science/r/12936/eZVx.pdf.
>
> ## W1. Frequency of Inaccurate 2-Step Priors
> We agree that validating the accuracy of the 2-step prior is essential. We conducted a comprehensive per-scenario analysis across all 66 Physics-IQ scenarios, measuring spatiotemporal IoU as it most directly captures motion dynamics (`Fig.1`, `Table1`) The prior improved motion dynamics in 74% (49/66) of scenarios for Wan 2.1 and 67% (44/66) for CogVideoX-5b. Critically, degradation is both infrequent and bounded: the maximum degradation (−20.3 / −22.2) is less than half the maximum improvement (+55.4 / +57.9), confirming that PhaseLock degrades gracefully rather than catastrophically. Our adaptive design (soft guidance λ₀=0.05 with linear decay) ensures that even when the prior is less informative, it does not override the base model's generation. The per-category breakdown (`Table 2, 3`) shows complementary strengths across models (Wan 2.1: 93% fluid dynamics; CogVideoX: 88% optics). We will add these to Appendix G.1.
>
> ## W2. Theoretical Justification
> We appreciate the opportunity to clarify the relationship between our theoretical analysis and the method's actual behavior. The equal-magnitude assumption is a sufficient condition for the closed-form in Eq. 5, not a necessary condition for the method. The general-case derivation (Sec. C) confirms that phase difference remains the dominant factor even with unequal magnitudes. Moreover, our spatial-domain L2 constraint aggregates over all frequencies via Parseval's theorem, requiring no individual frequency to satisfy this condition, which is precisely why our approach succeeds where direct frequency manipulation catastrophically fails (Table 8: baselines drop to 11–14; ours achieves 36.0).
>
> Our evaluation covers substantially dynamic content far beyond smooth motion. Among the most physically challenging scenarios in Physics-IQ,  involving fluid displacement (potato-in-water, +18.3), rapid deformation (cut-orange, +51.9), and capillary flow (napkin-soak, +54.7),  PhaseLock yields large improvements despite these involving abrupt velocity changes and discontinuous dynamics. Across all three backbone models, improvements average +6.2 points, and on PhyGenBench, all four categories improve including Mechanics (+13.3%/+11.6%) and Thermal (+11.9%/+28.9%), which inherently involve rapid state changes. The remaining performance ceiling is determined by the base model's own physical understanding, not by our method's assumptions, our adaptive guidance ensures that when the prior is less informative, it simply has less effect rather than causing harm. We will highlight the general-case derivation in the camera-ready that the equal-magnitude assumption serves as an analytical tool.
>
> ## Q1. The Effect of $K_few$
> We observe this monotonically decreasing trend consistently across models, though the rate of decline varies by architecture. Both WAN (28.7 → 28.4 → 27.9 → 26.2 → 24.2) and CogVideoX (36.0 → 32.8 → 30.5) show monotonic degradation as $K_few$ increases from 2 to 10, consistent with our phase erosion analysis where corruption accumulates at each denoising step. This confirms that 2-steps is the optimal choice as it captures motion priors before significant phase erosion occurs, regardless of the backbone. We will include in the camera-ready version.
>
> ## Q2. VAE OOD on Blurry Frames
> We clarify that the VAE encoder is not applied to blurry 2-step frames in our pipeline. As shown in Algorithm 1 (Sec. D.1), the entire motion prior extraction operates in latent space; the coarse latent $z^{few}$ is produced by the denoising model $\epsilon_\theta$, not by encoding pixel-space frames through the VAE. The VAE encoder is used only once to encode the high-quality reference image, and the decoder is applied only to the final 50-step output. Therefore, we respectfully clarify that the OOD concern does not apply, since the 2-step latent is simply an earlier step in the normal denoising process.
>
> ## Q3. Training-Based Physics Incorporation
> We agree that training-time physics incorporation is a compelling direction, discussed as key future work in both our Conclusion and Sec.G. In particular, our analysis reveals a concrete mechanism for why current training fails to preserve physics: the MSE objective's phase gradient is modulated by magnitude, causing it to vanish in high-frequency regions critical for motion dynamics. This finding directly suggests that phase-aware training losses could mitigate the erosion at its source. We believe our analysis, identifying phase erosion as the root cause and characterizing its relationship to the training objective, provides the necessary groundwork for designing such physics-aware training procedures in future work.

---

> > ### Author Rebuttal · Reviewer_eZVx · 2026-04-04
> >
> > Thank you to the authors for their detailed rebuttal, which has resolved my concerns. As for my question regarding VAE OOD, I would recommend authors to modify their overall pipeline figure in Figure 4, as it depicts blurry frames being input into the VAE which is misleading. I will maintain my score of 4.

---

> > > ### Author Response · Authors · 2026-04-07
> > >
> > > We are glad to have successfully addressed your concerns. We truly appreciate the constructive feedback throughout the review process, as your insightful feedback has made our work more solid.
> > >
> > > Regarding your recommendation on the VAE OOD issue, your point is well taken, and we completely agree. We will modify the pipeline in Figure 4 to avoid any potential misinterpretation.
> > >
> > > All the new analyses and experiments will be incorporated into the final version of the manuscript.

---

### Decision · Program_Chairs · 2026-04-30

**Decision:**

Accept (regular)

**Comment:**

This paper presents a training-free framework, PhaseLock, to address the "phase erosion" phenomenon—video diffusion models lose physical consistency during extended visual refinement. Main strengths include its impressive performance gains with minimal computational overhead (1.06x time) and its broad architectural generalizability across DiT and UNet models. Reviewers praised the rigorous spectral analysis and the novelty of using 2-step fast-inference priors to guide high-fidelity generation. Weaknesses initially centered on the robustness of the 2-step prior and potential over-constraining of visual details. However, the authors successfully mitigated these concerns by demonstrating graceful degradation (even with flawed priors), proving the method's efficacy in non-rigid dynamics (+41.8% gain), and clarifying that their Latent Delta Guidance functions as a soft nudge rather than a rigid lock. Consequently, all reviewers moved to a positive acceptance recommendation.